

# SimCloud version 1.0: a simple diagnostic cloud scheme for idealized climate models

Qun Liu[1], Matthew Collins[1], Penelope Maher[1], Stephen I. Thomson[1], and Geoffrey K. Vallis[1]

[1]College of Engineering, Mathematics and Physical Sciences, University of Exeter, Exeter, UK

**Correspondence:** Qun Liu (ql260@exeter.ac.uk)

**Abstract.** SimCloud, a simple diagnostic cloud scheme for general circulation models (GCMs) is proposed in this study. The large-scale clouds, which form the core of the scheme, are diagnosed from relative humidity. In addition, marine low stratus clouds, typically found off the west coast of continents over subtropical oceans, are determined largely as a function of inversion strength. A 'freeze-dry' adjustment based on a simple function of relative humidity may also used to reduce an
excessive clouds bias in polar regions. Other cloud properties, such as the effective radius of cloud droplet and cloud liquid water content, are specified as simple functions of temperature. All of these features are user-configurable. The cloud scheme is implemented in Isca, a modeling framework designed to enable the construction of GCMs at varying levels of complexity, but could readily be adapted to other GCMs. Simulations using the scheme with realistic continents generally capture the observed structure of cloud fraction and cloud radiative effect (CRE), as well as its seasonal variation. Specifically, the explicit low
cloud scheme improves the simulation of shortwave CREs over the eastern subtropical oceans by increasing the cloud fraction and cloud water path over there. The freeze-dry adjustment alleviates the longwave CRE biases in polar regions especially in winter. However, the longwave CRE in tropical regions and shortwave CRE over extratropics are still too strong compared to observations. Nevertheless, this simple cloud scheme provides a suitable basis for examining the impacts of clouds on climate in idealized modeling frameworks.

# 1 Introduction

Clouds usually cover more than half the area of the Earth at any given time (Houze, 2014) and play a fundamental role in the radiation budget of Earth. As is well known, clouds reflect the incoming shortwave (SW) radiation, so cooling the climate system, but at the same time they warm the system by absorbing the longwave (LW) radiation emitted from the surface (Hartmann, 1994). Previous studies have shown that the global mean net cloud radiative effect (CRE) is about -18 Wm$^{-2}$
(meaning that they cool the system), which is roughly five times as large as the radiative forcing of doubling $CO_2$ (e.g., Zelinka et al., 2017; Ramanathan et al., 1989; Loeb et al., 2009). Small changes in clouds can have a large impact on the climate energy balance due to balance between longwave and shortwave cloud effects. As a consequence, and not surprisingly, there are many studies focusing on the role of clouds under global warming – see, for example, Zelinka et al. (2017) and references therein.

    In spite of these studies, many issues about clouds and their interaction with the large-scale circulation remain unresolved.
The representation of clouds in General Circulation Models (GCMs) has become very complicated, with models differing from





one another both in the representation and in their results. As a result the net global mean cloud feedback remains uncertain and our best estimates are between -0.2 to 2 $\mathrm{Wm}^{-2}\mathrm{K}^{-1}$, as shown in the fifth assessment report of the Intergovernmental Panel on Climate Change (Stocker et al., 2013; Zelinka et al., 2017). Indeed cloud feedback is often thought to be the largest source of intermodel spread in equilibrium climate sensitivity (e.g., Ceppi et al., 2017), and considerable effort has gone into investigating

the possible causes for the large inter-model uncertainties in cloud feedbacks. Thus, for example, Webb et al. (2015) developed the Selected Process On/Off Klima Intercomparison Experiment (SPOOKIE) finding that the inter-model spread was not, in fact, due to parameterized convection. It was also shown using the SPOOKIE data that the model climatologies were similar in simulations with and without parameterised convection but the rainfall was unrealistic at daily timescales (Maher et al., 2018). The simulation of low clouds nevertheless remains problematic in GCMs: there are positive biases of the cloud radiative

forcing in marine stratus clouds regions (Dolinar et al., 2014), and there is little model agreement on low cloud amount change in the future among models in the fifth phase of the Coupled Model Intercomparison Project (Qu et al., 2013). With so many processes coupled together in GCMs it is not easy to understand the physical mechanisms behind global cloud feedbacks, and it is perhaps not surprising that results can differ considerably.

Perhaps the simplest recipe of representing clouds is to prescribe them with climatological data, without dynamic interplay
with the other components of the model, as in Holloway and Manabe (1971). A step up from this follows by noting that total water within a grid box follows some distributions, so that partial regions within it are saturated even when the grid box, on average, is not. Since clouds normally form on saturation then, depending on the form of the distribution, the cloud amount will be some function of mean relative humidity. A linear relationship between the non-precipitating cloud amount and relative humidity was adopted in early studies (e.g., Smagorinsky, 1960; Ricketts, 1973), and remains of considerable value, although

it certainly has its limitations (e.g., Ming and Held, 2018). More recent relative humidity schemes usually assume the cloud forms only when the grid mean relative humidity is larger than a critical relative humidity (e.g., Sundqvist et al., 1989; Slingo, 1980, 2007). In these schemes the critical relative humidity is usually determined empirically and may be a function of grid box size, and the cloud fraction and feedback may be rather sensitive to these threshold values (Quaas, 2012). Relatively simple diagnostic schemes are in fact still used in some comprehensive GCMs (e.g., Giorgetta et al., 2018).

In an attempt to move beyond such schemes, various more-or-less complicated prognostic and/or statistical cloud schemes have recently been widely employed in GCMs. The prognostic approach is to explicitly calculate the cloud-related variables (e.g. cloud water content) based on associated physical processes that constitute to the source and sink terms in the prognostic equations (e.g., Tiedtke, 1993). Statistical cloud schemes calculate the cloud fraction and condensate content consistently once the sub-grid probability density functions (PDFs) of certain variables such as total water specific humidity are determined

(e.g., Sommeria and Deardorff, 1977; Smith, 1990; Tompkins, 2002; Park et al., 2014; Qin et al., 2018; Tsang and Vallis, 2018). The proliferation of different cloud schemes, and their interaction with other parameterization schemes and the resolved dynamical flow, means that it is often very difficult to isolate the role of clouds in studies of climate variability and change. For this reason, we take a step back toward simplicity: our intent is to construct a relatively simple cloud scheme that can capture the key processes giving rise to clouds, and that enables us to better understand both their present-day geographical

distribution and their possible future change. We also seek to understand what might be a minimal recipe for reproducing cloud





effects and their variation in the atmosphere, and just what the limitations are of a scheme based solely on relative humidity. A complementary goal of the study is to develop a cloud scheme that can be used in GCMs, without the full complexity of a prognostic or statistical scheme, for more general climate studies.

The question then arises as to what a 'simple' scheme is. One option would be to specify the PDF of total water within
a grid box. Then, supposing that cloud formation occurs on saturation, one may be able derive a functional relation between mean cloud amount and mean relative humidity, supposing that the latter is what is predicted by the GCM from its predictions of specific humidity and temperature. Sundqvist et al. (1989) scheme was motivated this way, where the uniform distribution is adopted and variance of the distribution is assumed to be time-invariant (Tompkins, 2005). Although such a procedure is physically motivated it has two potential drawbacks. First, deciding on a distribution of humidity is somewhat arbitrary, or
involves turbulence closure assumptions used in the stochastic model (Sommeria and Deardorff, 1977; Tsang and Vallis, 2018). Second, translating the prediction of a probability distribution into a practical cloud model may be problematic, for there is in general no straightforward translation from a humidity probability distribution to an analytic formula connecting fractional cloud cover to relative humidity.

Thus, here we chose another course by linking the cloud cover with the relative humidity directly with simple forms. We
explore two schemes, one with a piecewise linear relationship between cloud cover and relative humidity and the other with a square-root relationship, as in Sundqvist et al. (1989). The various coefficients entering into these schemes are obtained empirically, comparing results with observations. We also use the scheme in an idealized GCM, Isca (Vallis et al., 2018), configured with a realistic distribution of continents to explore the geographical variability of the cloud schemes. Idealized models have a number of advantages in investigating physical processes, especially when set within a hierarchy connecting
them to more comprehensive models (e.g., Maher et al., 2019; Thomson and Vallis, 2019). We find that a relative-humidity scheme alone is unable to capture the subtropical low cloud distribution but that this can be readily improved by the addition of a scheme that takes into account inversion strength. Similarly, we find that in high-latitudes the cloud radiative effect is improved by the addition of a 'freeze-dry' scheme.

In its most complete form, the scheme is able to capture the key features (in both geographical and seasonal variability)
of observed clouds without the complexity of contemporary cloud schemes. It does so in a very transparent fashion and the dependence on parameters can be made explicit. And although we have implemented the scheme in a particular GCM it could easily be ported to others, either by a straightforward implementation or by porting the code itself.

We organize the paper as follows. Section 2 provides a description of the simple cloud scheme, SimCloud, including the methods to parameterize the cloud fraction and specify other needed physical parameters, such as the effective radius of
cloud droplets and in-cloud condensate content. (Some of the choices and parameters used come from the experiments and observations described in later sections, but for clarity the cloud scheme is described first.) Section 3 describes the model and experimental configurations as well as the data sets used in this study. In Sect. 4 we compare the simulated cloud properties with observations, with an emphasis on the CRE. A discussion and conclusions are presented in Sect. 5.



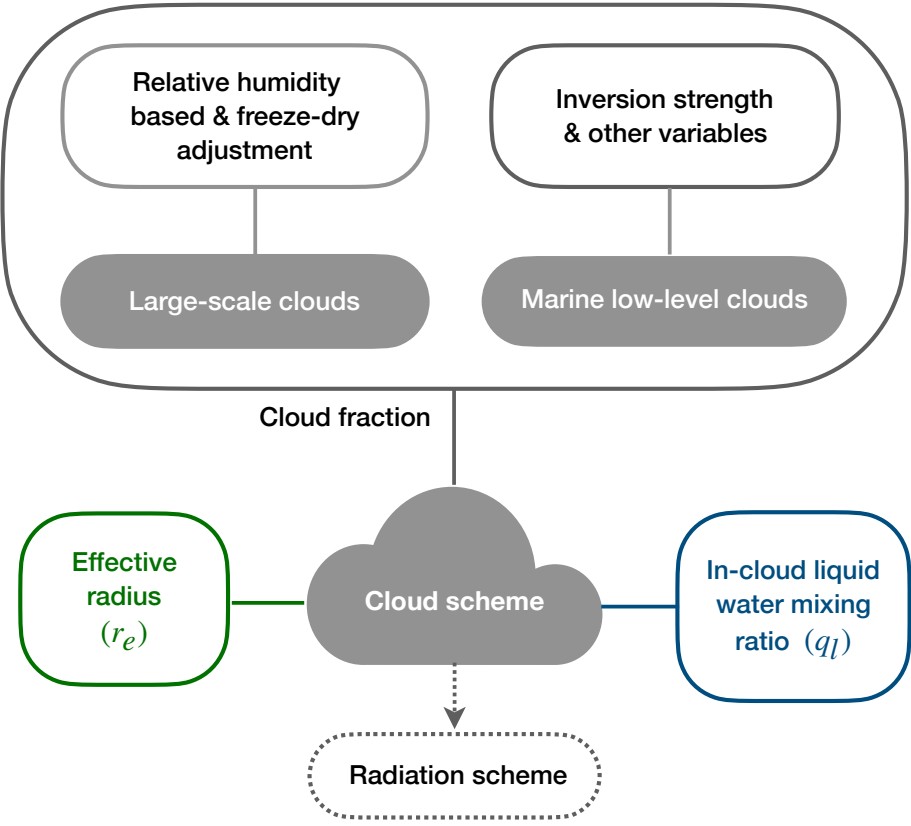

**Figure 1.** A sketch of the SimCloud scheme components, which include the cloud fraction, effective radius of cloud droplet and in-cloud liquid water mixing ratio. At any given location, the maximum of the cloud fractions from large-scale cloud scheme and marine low stratocumulus cloud scheme is applied if both of them are used.

## 2 SimCloud scheme description

### 2.1 Overview

In order to have a cloud scheme that interacts with the radiation, we need to predict not only the cloud amount but also its radiative properties. We focus mainly on the former, for the latter we require effective radius of the cloud droplets, and in-cloud liquid water content. In the following subsections we describe how these are specified; an encapsulation of the SimCloud scheme is also given in Fig. 1.

As noted in the introduction, large-scale clouds are parameterized as a function of relative humidity and this provides the majority of the cloud cover. However, as discussed in Sect. 4, this scheme alone is found to be inadequate and two additional effects are needed. First, a 'freeze-dry' method based on the specific humidity is used to reduce the large-scale cloud cover over





polar regions to more realistic levels. Second, a separate marine low stratus cloud scheme is used to represent the stratiform clouds (which have a large shortwave radiative effect), and this has a particularly large effect in subtropical regions off the

west coast of continents. These two additional components are optional and users can decide whether to use according to their research interest. Although these clouds have different physical properties (e.g., cloud top temperature), all of them are treated essentially as liquid clouds in our scheme. The effective radius of the cloud droplets is allowed to change with temperature, and this affects the radiative transfer. Some tuning of the clouds scheme is performed in order to fit the observations. Nevertheless, the values of certain parameters used in the scheme are not necessarily definitive and may be varied in order to examine the

sensitivity of clouds to perturbations such as $CO_2$ increase.

The present version does not include a separate scheme for convective clouds, and the convection scheme in the model has no effect on cloudiness except in so far as it may change the relative humidity or, possibly, the low-level inversion. We find that the vertical structure of clouds can be simulated relatively well without explicit diagnosis of convective clouds (see Fig. 6), and we leave the possible explicit representation of convective clouds to a future study.

## 2.2 Cloud fraction

### 2.2.1 Relative humidity-based cloud fraction

The use of a relative humidity scheme is based on the notion that over a grid box the humidity varies, and that condensation will occur and clouds will form even when the grid cell is not saturated (Tompkins, 2005; Quaas, 2012), and one such scheme is that of Sundqvist et al. (1989), discussed more below. Such schemes do not account for variations in dynamical conditions

except in so far as they are reflected in the relative humidity field, but they form a simple rational basis for cloud prediction. Here we implement two relative humidity schemes with different forms to diagnose the cloud fraction.

In the first scheme, the large-scale cloud fraction ($C_s$) is a piecewise linear function of grid-mean relative humidity ($H$), namely

$$C_s = \min\left(1,\ \max\left(0,\ a \cdot (H-1) + 1\right)\right). \tag{1}$$

The diagnosed cloud fraction is therefore unity when the mean relative humidity equal to one (i.e. grid-box is saturated). The value of $a$ determines the critical value of relative humidity, $H_c$, above which clouds form, so that $a = 1/(1 - H_c)$ and $H_c = (a-1)/a$. The coefficient $a$ (and hence $H_c$) is taken to be a function of height (or pressure) but not latitude or longitude.

The vertical profile of the coefficient $a$ was derived from the reanalysis data sets. Specifically, the hourly relative humidity and cloud fraction data sets in year 2017 from the European Centre for Medium-Range Weather Forecasts (ECMWF) Reanal-

ysis version 5 (ERA5) (Copernicus Climate Change Service (C3S), 2017) are used to derive the coefficient profile. The ERA5 has $1° \times 1°$ horizontal resolution and 37 vertical levels. For each vertical level, the piecewise linear relationship (as cloud fraction is not allowed to be smaller than 0 or larger than 1) is used to fit the cloud fraction against relative humidity with a least squares method, then the coefficient $a$ of that level is obtained. In addition, we have re-gridded the data to three other horizontal resolutions, including $0.75° \times 0.75°$, $1.4° \times 1.4°$ (T85) and $2.8° \times 2.8°$ (T42), to test whether the derived coefficient profile is

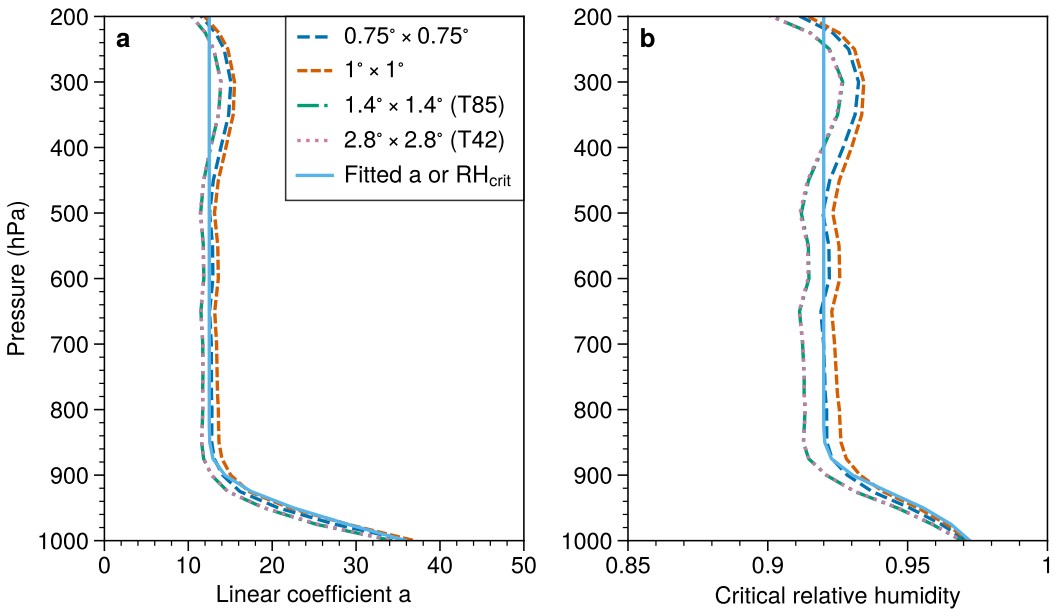

**Figure 2.** (a) Vertical profiles of the linear coefficient for the relative humidity-based cloud diagnostic scheme, with cloud fraction as a piecewise linear function of relative humidity as shown in Equation (1). The dashed lines with different colors are profiles of $a$ obtained from ERA5 data with various horizontal resolutions. The solid pale blue line denotes the fitted profile used in the model based on the form of Equation (2). (b) is the same as (a), but for the critical relative humidity ($H_c$) profiles.

sensitive to horizontal resolution. The derived profiles with different resolutions are shown in Fig. 2a, and the corresponding critical relative humidity ($H_c$) profiles are shown in Fig. 2b for reference. At very high resolution we would expect the humidity distribution in a given grid box to be narrower, and hence the critical value of relative humidity to increase, however we find that the horizontal resolution has a relatively small influence on the coefficient profiles at the resolutions we consider here.

To apply the coefficient profile of $a$ in the model, a profile similar to the Equation (3) from Quaas (2012) is used:

$$a = a_t + (a_s - a_t) \exp \left[ 1 - \left( \frac{p_s}{p} \right)^n \right], \tag{2}$$

where $p$ is pressure, $p_s$ is the surface pressure, $a_s$ and $a_t$ are the values of coefficient $a$ at surface and free troposphere respectively. Such a functional form fits the observations fairly well with only a small number of tunable parameters. We use $a_s = 36$, $a_t = 13$ and $n = 12$, which determines shape of the profile. The fitted profile for $a$, as indicated by the solid pale blue line in Fig. 2, follows the reanalysis (dashed/dotted lines) quite well at low and middle levels but with some discrepancies

at higher levels. The actual cloud fraction for each level is determined by Equation (1) with the coefficient $a$ for that level determined by Equation (2).

We also provide the Sundqvist et al. (1989) scheme as another option for the relative-humidity schemes, namely

$$C_s = 1 - \sqrt{\frac{1 - H}{1 - H_c}}, \tag{3}$$





where $H_c$ is critical relative humidity. Here we specify $H_c$ as a simple function of height, which is determined by critical

relative humidity at three different levels: 0.95 at the surface, 0.85 at 700 hPa, and 0.99 at 200 hPa. Between these levels, the critical relative humidity is linearly interpolated with height. To test the performance of the aforementioned schemes, we compare them with another linear scheme with a different form from Equation (1), which is defined as

$$C_s = \min\left(1, \, \max\left(0, \, a \cdot H + b\right)\right), \tag{4}$$

where $a$ and $b$ are determined from the least squares fitting of hourly cloud fraction and relative humidity data from ERA5

reanalysis. The cloud fraction in Fig. 3a is from ERA5 reanalysis at 450 hPa on 12:00 January 01, 2017, while the cloud fractions from three schemes (Figs. 3b-3d) are diagnosed from the ERA5 relative humidity field at the same time and level. The linear scheme defined in Equation (4) has two tunable parameters, so one might expect it to perform better than the others. However, the cloud cover can not reach 1 when the grid box is saturated (Fig. 3c), even though the spatial pattern of cloud cover resembles the ERA5 reanalysis and the global mean value is much closer to the ERA5 compared to the other two schemes. In

contrast, the diagnosed cloud amount patterns from Sundqvist scheme (Fig. 3b) and the linear scheme in the form of Equation (1) (Fig. 3d) are quite similar to the reanalysis (Fig. 3a), although the cloud cover is a little overestimated in these two schemes. The offline test suggests that the linear scheme in the form of Equation (1) is promising to be applied in GCMs.

### 2.2.2    Freeze-dry adjustment

As we will show in Sect. 4.1, there are biases in the cloud fraction and cloud radiative effect from the large scale cloud scheme

in the polar regions, especially during winter. While it is common for the relative humidity to be near saturation in the Arctic, especially during winter, Jones et al. (2004) showed that there is little subgrid-scale heterogeneity of relative humidity in these stable environments. The small quantity of condensation nuclei in this region further limits the formation of clouds. To alleviate this problem, the 'freeze-dry' adjustment, a simple adjustment formula based on specific humidity as indicated by the Equation (2) from Vavrus and Waliser (2008), is applied to reduce the cloud fraction under very dry conditions in polar regions.

Specifically, if grid mean specific humidity ($q$) is below a threshold ($q_v$), the cloud fraction ($C$) decreases linearly according to the water vapor content:

$$C = C \cdot f = C \times \max\left(0.15, \min\left(1.0, \, \frac{q}{q_v}\right)\right), \tag{5}$$

where $f$ is referred to as the freeze-dry factor. Although the formula is applied globally, the threshold value in Equation (5) ensures that only polar regions will be affected and even there the cloud fraction is adjusted only under very dry conditions.

In the original freeze-dry method, Equation (5) was only applied in the lower troposphere (Vavrus and Waliser, 2008). In this study, the freeze-dry formula is applied through the whole atmospheric column, finding that the cloud radiative effect in polar regions is thereby improved. In order to do so, we prescribe the specific humidity threshold $q_v$ in Equation (5) to be a function of pressure with the threshold decreasing exponentially with height as

$$q_v = q_0 \left(\frac{p}{p_s}\right)^n. \tag{6}$$





**Figure 3.** (a) A snapshot of cloud fraction from ERA5 reanalysis at 450 hPa on 12:00 January 01, 2017. Diagnosed cloud fraction from ERA5 relative humidity field at the same time and level based on the (b) Sundqvist formula, and two linear formulas (c) using Equation (4) and (d) using Equation (1). Note that Equation (1) is the form used to determine the large-scale clouds in this study. The global mean cloud fractions are given in the titles.



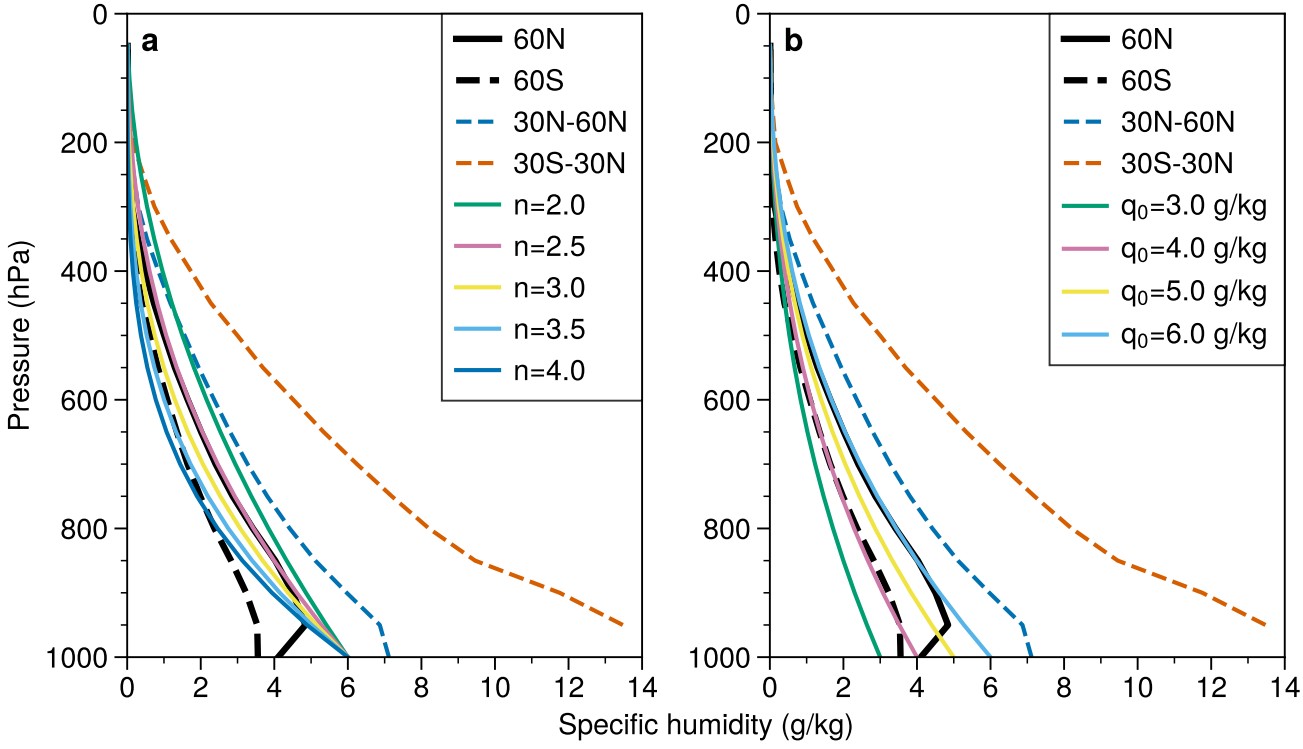

**Figure 4.** The $q_v$ vertical profiles with different $n$ and $q_0$. The thick solid and dashed black lines are specific humidity profiles averaged over latitude circles, 60N and 60S, as boundary values of polar regions in Northern and Southern hemispheres, respectively. The thin dashed blue and orange lines are averaged specific humidity profiles over subtropical ($30°$-$60°$N) and tropical ($30°$S-$30°$N) regions in Isca simulation. The other thin solid lines are from Equation (6) with different parameters. In the left, $q_0$ in Equation (6) is 0.006 kg kg$^{-1}$ but $n$ varies from 2 to 4. In the right, $n = 2.5$ but $q_0$ varies from 0.003 to 0.006 kg kg$^{-1}$.

Here $q_0$ is the surface specific humidity, $p_s = 1000$ hPa is the sea level pressure and $n$ is the power to describe how quickly the specific humidity decreases with height. In Fig. 4, different profiles of $q_v$ are shown for the two tunable parameters $n$ and $q_0$. These two parameters are selected to ensure that the freeze-dry adjustment only has effects on polar regions when the $q_v$ profile is applied in Equation (5). In doing so, the specific humidity profiles from several different regions are plotted in Fig. 4. In particular, the profiles of $60°$N and $60°$S are used to show the specific humidity boundary values of polar regions, and thus

the two parameters $q_0$ and $n$ are tuned to follow the boundary profiles. As shown in Fig. 4, the $q_v$ profile follows $60°$N profile well when the $q_0 = 0.006$ kg kg$^{-1}$ and $n = 2.5$, which can also cover the specific humidity range poleward of $60°$S. Therefore in this study the parameters $q_0$ and $n$ are chosen as 0.006 kg kg$^{-1}$ and 2.5, respectively. This threshold works well in current climate setup (see Sect. 4.1), but whether it holds under global warming situation still needs further investigation.



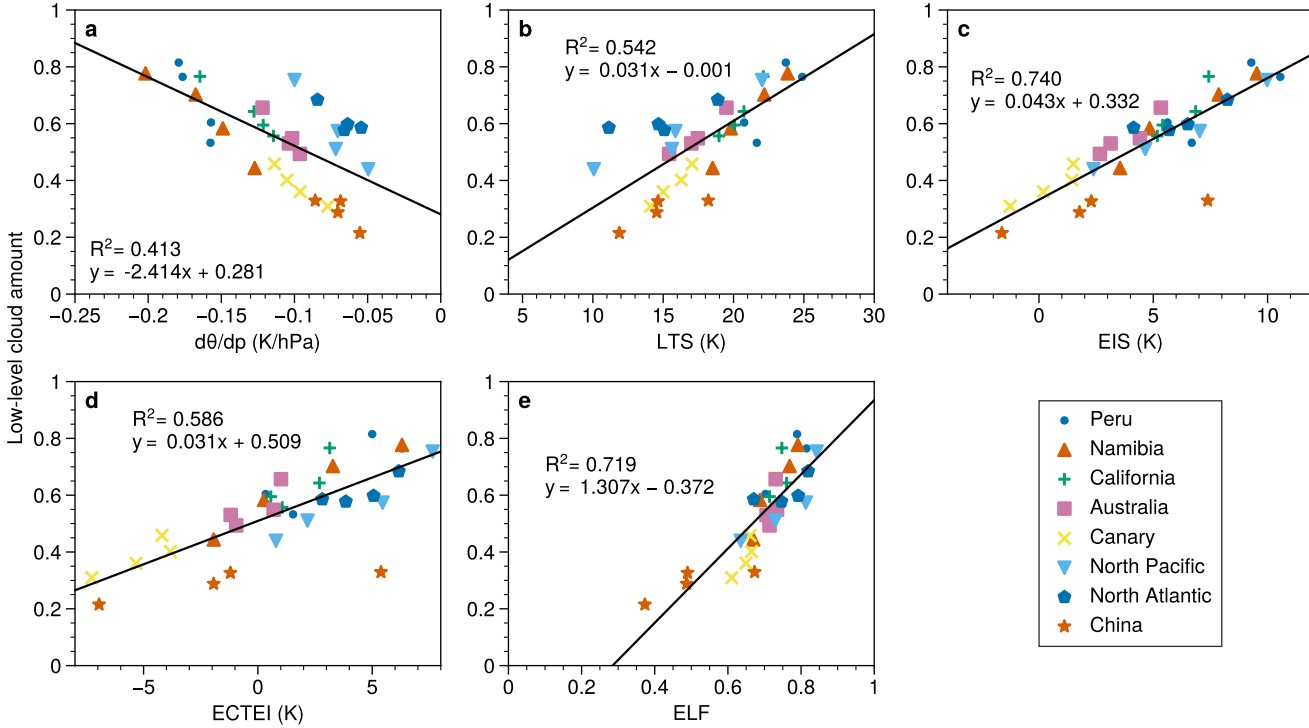

**Figure 5.** The relationship between low-level cloud amount and (a) minimum $d\theta/dp$ below 750 hPa, (b) lower tropospheric stability (LTS), (c) estimated inversion strength (EIS), (d) estimated cloud-top entrainment index (ECTEI) and (e) estimated low cloud fraction (ELF) over stratiform cloud regions, including Peru (20°S-5°N, 80°-90°W), Namibia (10°-30°S, 0°-15°E), California (15°-30°N, 110°-150°W), Australia (15°-35°S, 90°-110°E), Canary (15°-25°N, 25°-35°W), North Pacific (40°-5°0N, 170°-180°E), North Atlantic (50°-60°N, 35°-45°W) and China (20°-30°N, 105°-120°E), which are selected based on Klein and Hartmann (1993). The data sets are from ERA-Interim reanalysis covering the period from 2013 to 2017. The four points in each region denote the average for different seasons. Linear regression lines and the corresponding fraction of variance ($R^2$) explained by the equation are shown at the top of each plot.

### 2.2.3 Low cloud fraction

Low clouds, especially the subtropical marine stratocumulus clouds, are characterized by high albedo and a cooling effect on climate (Hartmann et al., 1992). Because these clouds cover about 20% of the subtropical regions even a small change in stratocumulus cloud amount can exert a large radiative forcing at the top of the atmosphere (TOA) (Slingo, 1990). However, marine stratocumulus amounts off the west coast of continents have commonly been underestimated and the been an issue in climate models for some time (e.g., Nam et al., 2012; Lauer and Hamilton, 2013; Dolinar et al., 2014).

Several proxies or indices for low cloud fraction have been used as predictors for the stratocumulus clouds to try to remedy this (e.g., Kawai and Inoue, 2006; Joshi et al., 2015; Collins et al., 2004; Guo and Zhou, 2014; Kawai et al., 2019), including potential temperature lapse rate ($d\theta/dp$) of the most stable layer below 750hPa (Slingo, 2007); lower tropospheric stability



(LTS; Klein and Hartmann, 1993); estimated inversion strength (EIS; Wood and Bretherton, 2006); and the estimated cloud-top entrainment index (ECTEI; Kawai et al., 2017). Recently, Park and Shin (2019) proposed a new index, the estimated low-level cloud fraction (ELF), as a predictor for low cloud fraction. ELF (which is a proxy and not necessarily a cloud amount itself) is defined as

$$\text{ELF} \equiv f \cdot \left[ 1 - \frac{\sqrt{z_{inv} \cdot z_{LCL}}}{\Delta z_s} \right], \tag{7}$$

where $f$ is the freeze-dry factor defined in (5) with $q_v = 0.003$ kg kg$^{-1}$ and $q$ is the surface water vapor specific humidity, $z_{inv}$ is the inversion height, $z_{LCL}$ is the lifting condensation level of near-surface air, and $\Delta z_s = 2750$ m is a constant scale height. As pointed by Park and Shin (2019), $\sqrt{z_{inv} \cdot z_{LCL}}/\Delta z_s$ can be rewritten as $z_{LCL}/\Delta z_s \cdot \sqrt{1 + (z_{inv} - z_{LCL})/z_{LCL}}$, in which $z_{LCL}/\Delta z_s$ is a simple but practical proxy of surface moisture, and $(z_{inv} - z_{LCL})/z_{LCL}$ quantifies the strength of the vertical decoupling of the inversion base air from the surface. The ELF predicts that low-level cloud fraction increases as the near-surface air gets more wet (smaller $z_{LCL}$) and as the planetary boundary layer becomes more vertically coupled (smaller $z_{inv}$).

We have examined the relationship between the seasonal mean low cloud fraction and the various proxies (i.e., $d\theta/dp$, LTS, EIS, ECTEI and ELF) using the ERA-Interim reanalysis data set (Dee et al., 2011). These proxies are derived from the five-year monthly data from 2013 to 2017, including air temperature, surface pressure, surface temperature and low cloud fraction. As shown in Fig. 5, the regions with typical stratus clouds are selected for the calculation (Klein and Hartmann, 1993). The results indicate that the low cloud fraction is linearly related to each indicator in stratus clouds regions, and the ELF tends to have very high correlation with the low-level cloud cover, judging from the fraction of variance ($R^2$) explained by the regression equation. We thus choose to use ELF to construct the diagnostic low cloud fraction formula, that is:

$$C_{sc} = \min(1, \ \max(0, \ b \times \text{ELF} + c)), \tag{8}$$

where $C_{sc}$ is the low stratus cloud fraction, and the two coefficients $b$ and $c$ are treated as tunable parameters. The linear regression formula displayed in Fig. 5e provides a good starting point for tuning $b$ and $c$ in Equation (8). After a sensitivity test performed with Isca (the setups will be introduced in Sect. 3), we find that if $c$ is specified as the value shown in Fig. 5e, the shortwave cloud radiative effect is still weak compared to observations. Therefore the parameters $b$ and $c$ are chosen as 1.3 and -0.1 respectively.

In addition, the stratocumulus clouds usually form at the top of planetary boundary layer (Wood, 2012), where a strong inversion layer usually exists (Wood and Bretherton, 2006; Park and Shin, 2019). However, it is hard for a global model to capture the exact position of the inversion layer due to the limitation of vertical resolution (Kawai et al., 2019). Care thus needs to be taken to diagnose the marine stratocumulus clouds. First we find the most stable layer below 750 hPa, which is determined by the most negative $d\theta/dp$ (Slingo, 2007). Then within the most stable layer, if the lapse rate and vertical velocity satisfy $d\theta/dp < -0.08$ K/hPa and $\omega < 0$ hPa/s respectively, then we diagnose stratocumulus clouds at that location. Note that the $d\theta/dp$ threshold is tuneable in our scheme, and it is $-0.125$ K/hPa, as in Collins et al. (2004).





### 2.2.4 Cloud fraction diagnosis


The cloud fraction of a grid box ($C_{\text{total}}$) is simply defined as the largest fraction of all the clouds within that grid box for simplicity, without separate consideration of their different optical properties:

$$C_{\text{total}} = \max(C_s, \, C_{sc}), \tag{9}$$

assuming a horizontal maximum overlap hypothesis (e.g., Collins et al., 2004; Roehrig et al., 2020). $C_s$ and $C_{sc}$ in Equation

(9) are determined by Equations (1) and (8) respectively. To assess the performance of the cloud scheme, it is useful to evaluate the total cloud amount and cloud amounts at different levels. In our scheme, the cloud height is determined by cloud top pressures, where those located above 400 hPa are treated as high clouds, those below 700 hPa are defined as low clouds, and in between are middle clouds (Collins et al., 2004). Then, the total, high, middle or low cloud amounts are diagnosed from the maximum-random overlap assumption (Morcrette and Jakob, 2000), which assumes maximum overlap for consecutive cloudy

model levels and random overlap for cloud layers that are separated by clear-sky levels.

### 2.3 Effective radius and in-cloud water mixing ratio

Cloud particles, including liquid droplets and ice crystals, usually have different sizes, shapes and optical properties. In order not to introduce complicated microphysical processes, we do not distinguish them and assume that all particles seen by the radiation scheme are spherical liquid droplets, and ice clouds have a different effective radius from the liquid ones. In this

study, the liquid cloud fraction varies with temperature, which only has an influence on the effective radius.

Following Ose (1993) and Boville et al. (2006), a very simple approach is used to represent the liquid cloud fraction ($f_l$) within a grid box. Specifically, all clouds are assumed to be in liquid form if temperature is warmer than $T_{max}$, and all the condensate is considered as ice if temperature is colder than $T_{min}$. The cloud droplets are in mixed-phase at temperatures between $T_{min}$ and $T_{max}$, and the proportion of liquid cloud in a grid box is defined as a linear function of temperature:

$$f_l = \max\left(0, \, \min\left(1, \, \frac{T - T_{min}}{T_{max} - T_{min}}\right)\right). \tag{10}$$

The bounds $T_{min}$ and $T_{max}$ are different in different models. For example, the lower bound ($T_{min}$) is $-40°$C in Ose (1993) and Boville et al. (2006), while it is $-15°$C in Smith (1990). Observation have shown that cloud liquid water can exist at temperature as low as $-40°$C (Heymsfield and Miloshevich, 1993), although the incidence of liquid water in stratiform clouds is quite low at temperatures below $-15°$C (Ryan, 1996). The upper bounds ($T_{max}$) are $-5°$C for stratiform clouds in Ose

(1993), $-10°$C in Boville et al. (2006), and $0°$C in Smith (1990). Based on the choices in previous studies, $T_{min}$ and $T_{max}$ in Equation (10) are chosen to be $-40°$C and $-5°$C respectively in this study, but they are to be regarded as adjustable parameters.

The effective radius ($r_e$) of droplets within a grid box is defined as a weighted mean of liquid and ice particle radii, with the weights given by the liquid and ice cloud fraction respectively. The radii of liquid and ice particles are selected based on observations. Stubenrauch et al. (2013) assessed cloud properties derived from various satellite data sets, finding that the

global mean effective particle radii are about 14 ($\pm1$) and 25 ($\pm2$) $\mu m$ for the tops of liquid clouds and for high-level ice



**Table 1.** Summary of the simple diagnostic cloud scheme. Note that two options are provided for the large-scale cloud fraction ($C_s$).

| Symbol | Range /Units | Definition | Diagnostic formula | Tunable parameters |
|---|---|---|---|---|
| $C_s$ | [0, 1] | Large-scale cloud fraction | $\min\left(1,\max\left(0,a\cdot(H-1)+1\right)\right),$ $a=a_t+(a_s-a_t)\exp\left[1-\left(\frac{p_s}{p}\right)^n\right]$ | $a_s$=36, $a_t$=13, $n$=12 |
| | | | $\min\left(1,\max\left(0,1-\sqrt{\frac{1-H}{1-H_c}}\right)\right)$ | $H_c$: function of height |
| $f$ | [0.15, 1] | Freeze-dry adjustment factor | $\max\left(0.15,\min\left(1.0,\frac{q}{q_v}\right)\right),$ $q_v=q_0\left(\frac{p}{p_s}\right)^n$ | $q_0$=6 g kg$^{-1}$, $n$=2.5 |
| $C_{sc}$ | [0, 1] | Low cloud fraction | $\min(1,\max(0,b\times \mathrm{ELF}+c)),$ $\mathrm{ELF}=f\cdot[1-\sqrt{z_{inv}\cdot z_{LCL}}/\Delta z_s]$ | $b$=1.3, $c$=$-0.1$ $q_v$=3 g kg$^{-1}$ in $f$ |
| $f_l$ | [0, 1] | Liquid cloud fraction | $\max\left(0,\min\left(1,\frac{T-T_{min}}{T_{max}-T_{min}}\right)\right),$ ($T$ in units of °C) | $T_{min}$=$-40$, $T_{max}$=$-5$°C |
| $r_e$ | $\mu$m | Effective radius | $r_{e\_liq}f_l+r_{e\_ice}(1-f_l)$ | $r_{e\_liq}$=14, $r_{e\_ice}$=25 $\mu$m |
| $w_l$ | g kg$^{-1}$ | In-cloud liquid water mixing ratio | $\max\left(3\times10^{-4},w_{l0}\times\min\left(1,\frac{T-220}{280-220}\right)\right),$ ($T$ in units of K) | $w_{l0}$=0.18 g kg$^{-1}$ |

clouds, respectively. Therefore these two values are selected to calculate $r_e$,

$$r_e = 14f_l + 25(1-f_l), \tag{11}$$

which is applied globally in the model, although the effective radius of cloud droplets is found a little larger over ocean than over continents in observations (Stubenrauch et al., 2013).

The in-cloud liquid water mixing ratio ($w_l$) is specified as a linear function of the atmospheric temperature, with values of $3\times10^{-4}$ g kg$^{-1}$ at 220 K and $w_{l0}=0.18$ g kg$^{-1}$ at 280 K:

$$w_l = \max\left(3\times10^{-4},\ w_{l0}\times\min\left(1,\frac{T-220}{280-220}\right)\right), \tag{12}$$

where the atmospheric temperature $T$ is in units of K. The temperature thresholds, 280 and 220 K, are selected close to the global averages of liquid and ice cloud top temperature in observations, respectively (Fig. 4 in Stubenrauch et al., 2013). Then

the grid mean liquid water specific humidity can be obtained from the product of $w_l$ and cloud fraction. For reference, the equations and parameters used in the cloud scheme are summarized in Table 1.

## 3    Experiments and data sets

The SimCloud scheme was implemented into Isca (Vallis et al., 2018) to examine its performance. Isca is an open-source framework for the construction of general circulation of atmospheres, which is built around a dynamical core from the Geo-

physical Fluid Dynamics Laboratory (GFDL) and physical parameterizations from Frierson et al. (2006) and Frierson (2007).



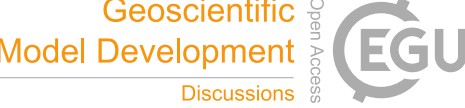

**Table 2.** Summary of the Isca fixed-SST simulations used in this study.

| Experiment | Description |
|---|---|
| LS | Run with large-scale clouds only. |
| FD | Based on the LS run, with a freeze-dry adjustment also applied. |
| ALL | The marine low-level clouds are also included on top of the FD run. |
| Linear_X | X is one of the LS, FD and ALL runs, in which the large-scale clouds are diagnosed from a linear function of RH as defined in (1). |
| Sundqvist_X | Same as Linear_X, but with Sundqvist et al. (1989) scheme as defined in (3). |

Isca provides various options for users to set up experiments for their own interests, which include dry and moist models, various convection and radiation schemes, a variety of land/sea configurations and different parameters for other planetary atmospheres.

Our simulations are AMIP-type, that is they follow those used in the Atmospheric Model Intercomparison Project. They are performed with a realistic Earth-continental configuration following Thomson and Vallis (2018) (which is derived from the ERA-Interim land mask and topography (Dee et al., 2011)) and at a horizontal resolution of T42 (roughly $2.8° \times 2.8°$) with 25 vertical levels. The sea surface temperature is fixed at AMIP climatology (Taylor et al., 2000), which is annually repeating but seasonally varying. The sea ice data is also from AMIP, but is averaged over all years and months to obtain an annual mean distribution, as in Thomson and Vallis (2018). The albedo in sea ice regions increases linearly with the sea ice concentration with the maximum of 0.7. The insolation includes a seasonal and diurnal cycle, with a solar constant of 1360 $Wm^{-2}$. The convection parameterization used in this study is the simplified Betts–Miller scheme from Frierson (2007). The SOCRATES (Suite of Community Radiation Codes based on Edwards & Slingo) radiation scheme (Edwards and Slingo, 1996; Manners et al., 2015) is employed for the radiation transfer calculation as in Thomson and Vallis (2019). Spectral files with 9 longwave bands and 6 shortwave bands are used, which are those used in the Unified Model's Global Atmosphere version 7 (Walters et al., 2019). The cloud fraction, effective radius of cloud particle and liquid water mixing ratio in each grid are passed to it, then the radiation fluxes under all-sky and clear-sky conditions are obtained, which are used to analyze the energy balance and to calculate the cloud radiative effect (Ramanathan et al., 1989; Li et al., 2017) at the TOA.

In order to compare the roles of different cloud parameterization schemes, simulations are performed with the combination of different clouds or different adjustment methods as shown in Table 2. The simulation with large-scale clouds only is denoted as the LS simulation. The run with large-scale clouds and freeze-dry adjustment is called the FD simulation. The run performed with large-scale clouds, freeze-dry adjustment and marine low stratiform clouds is referred as the ALL simulation. The simulations are all run for 20 years, with the first 10 years treated as spin-up and discarded.

To evaluate the performance of the cloud scheme, several observations and reanalysis data sets are employed. Specifically, the 10-year monthly data (1995 to 2014) of International Satellite Cloud Climatology Project (ISCCP) H-series products (Young et al., 2018) is used to evaluate the simulated cloud amounts (Sect. 4.1). The cloud fraction from Isca simulations is compared to retrieved cloud fraction from GCM-Oriented Cloud-Aerosol Lidar and Infrared Pathfinder Satellite Observations (CALIPSO)





**Table 3.** Global and annual mean climatological properties of observations and different Isca simulations, which are summarized in Table 2. The net fluxes in the table are positive downward. The observed cloud amounts are from ISCCP H-series product (1995-2014), the radiative fluxes and cloud radiative effects (CREs) at the TOA are from CERES-EBAF data set (2001-2018), and the cloud water path is from CloudSat data product.

|  | Obs | Linear_LS | Linear_FD | Linear_ALL | Sundqvist_LS | Sundqvist_FD | Sundqvist_ALL |
|---|---|---|---|---|---|---|---|
| Low cloud amount (%) | 28.0 | 54.6 | 49.3 | 48.6 | 53.8 | 48.3 | 47.5 |
| Middle cloud amount (%) | 20.8 | 25.9 | 20.8 | 20.6 | 25.3 | 20.2 | 20.0 |
| High cloud amount (%) | 12.8 | 42.3 | 30.4 | 30.4 | 35.9 | 25.5 | 25.5 |
| Total cloud amount (%) | 65.3 | 75.6 | 66.4 | 66.0 | 72.4 | 63.3 | 62.6 |
| TOA net SW flux (Wm$^{-2}$) | 241.3 | 226.8 | 229.2 | 229.6 | 228.7 | 231.2 | 231.5 |
| TOA net LW flux (Wm$^{-2}$) | 240.3 | 220.4 | 224.8 | 224.9 | 223.3 | 227.6 | 227.4 |
| TOA net flux (Wm$^{-2}$) | 1.0 | 6.4 | 4.5 | 4.7 | 5.4 | 3.6 | 4.0 |
| TOA SW CRE (Wm$^{-2}$) | -45.8 | -58.8 | -56.3 | -56.0 | -56.9 | -54.3 | -54.1 |
| TOA LW CRE (Wm$^{-2}$) | 28.0 | 36.4 | 31.3 | 31.0 | 33.3 | 28.5 | 28.3 |
| TOA net CRE (Wm$^{-2}$) | -17.8 | -22.4 | -25.0 | -24.9 | -23.5 | -25.8 | -25.8 |
| Cloud water path (gm$^{-2}$) | 119.3 | 147.0 | 129.7 | 130.4 | 144.1 | 127.1 | 126.8 |

Cloud Observations product (GOCCP) (Chepfer et al., 2010). To examine the radiative flux simulated in Isca, monthly data from January 2001 to December 2018 from Clouds and Earth's Radiant Energy System (CERES) Energy Balanced and Filled (EBAF) Edition 4.1 product (CERES-EBAF hereafter; Loeb et al., 2018) are used for comparison. The cloud water path is from
the CloudSat 2B-CWC-RO Release P1_R05 data product (Austin et al., 2009) from 2012 to 2016, which can better represent cloud liquid and ice water path over high latitudes than CERES-EBAF data set, owing to its explicit determination of cloud phase (Lenaerts et al., 2017). In addition, monthly vertical pressure velocity from ERA-Interim reanalysis and radiative flux data from CERES-EBAF data sets covering the period 2008-2017 are also adopted to quantify the LW CRE over the tropics.

In order to demonstrate how this cloud scheme performs with respect to more comprehensive models, the monthly mean
radiative fluxes at clear-sky and all-sky conditions in historical simulation (1996 to 2005) from various CMIP5 models are also shown (see Fig. 17 for the names of models). All the data sets are remapped to T42 resolution when necessary for a direct comparison with Isca simulations.

## 4 Results

The global mean cloud amount and radiative components for the observations and Isca simulations are summarized in Table 3.
We first focus on the mean state of the cloud amount, cloud water path and CRE at TOA from the LS, FD and ALL simulations (sections 4.1 to 4.3). Then, in Sect. 4.4 we compare the simulated CREs with CMIP5 models. The sensitivity of the cloud scheme to the choice of parameters is briefly presented in Sect. 4.5.



**Figure 6.** (a) The annual and global mean of cloud fraction profiles from the CALIPSO-GOCCP (thick blue solid line), ERA-Interim reanalysis (thick orange solid line) and different Isca simulations, including linear_LS (blue dotted), linear_FD (orange dash-dotted), linear_ALL (green dashed), Sundqvist_LS (pink dotted), Sundqvist_FD (yellow dash-dotted) and Sundqvist_ALL (azure dashed). (b-i) Same as in (a), but for annual and zonal mean of cloud fraction profiles.

## 4.1 Cloud amount



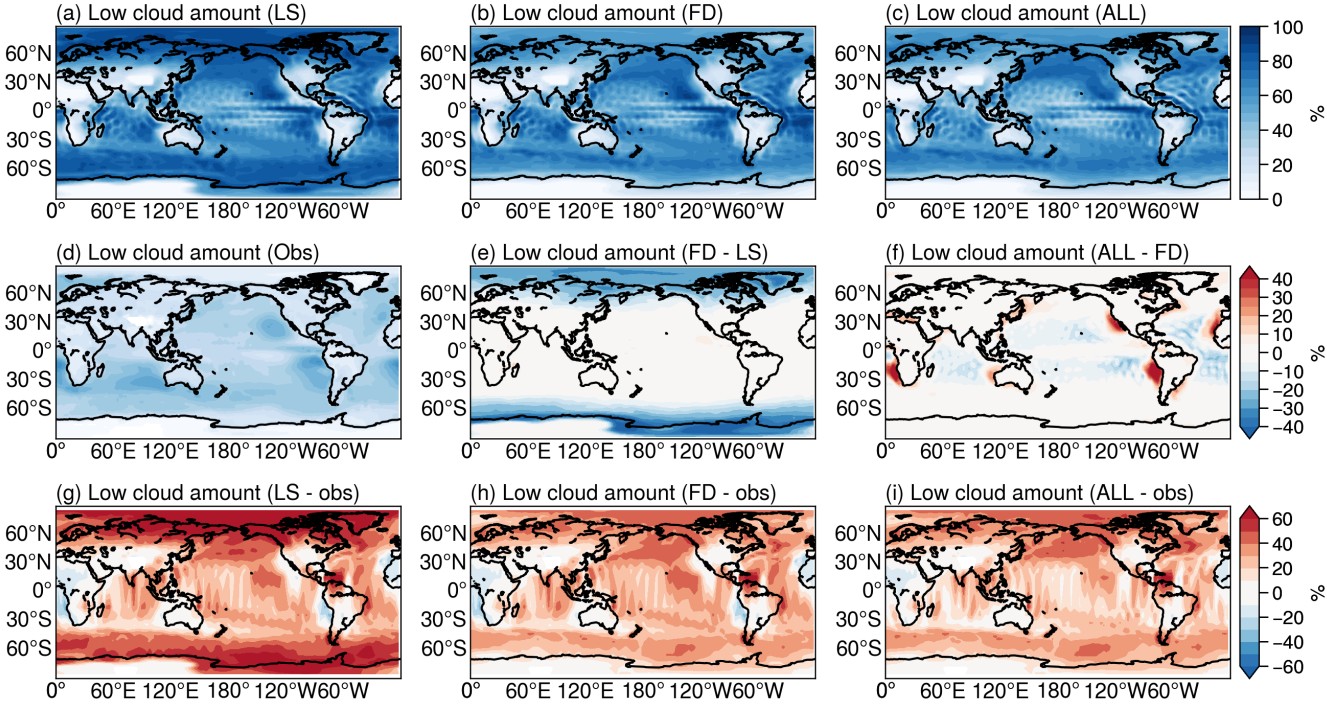

**Figure 7.** The annual mean geographic patterns of low cloud amount (%) from the (a) LS, (b) FD, (c) ALL simulations with the linear scheme as well as (d) observation (ISCCP-H), and the differences between the (e) FD and LS, (f) ALL and FD, (g) LS and observation, (h) FD and observation, and (i) ALL and observation. Note that (a-d) use the upper color scale, (e-f) use the middle one, and (g-i) use the one at the bottom.

The global mean cloud fraction profiles from CALIPSO-GOCCP, ERA-Interim reanalysis and Isca simulations are displayed
in Fig. 6a. The cloud fractions from all the Isca simulations are higher than observations, especially in the middle and high
levels. The FD simulations are closer to observations than the LS simulations, which is true for both the linear and Sundqvist
schemes. Regarding the annual and zonal mean profiles, a striking feature is that the LS simulations from both linear and
Sundqvist schemes overestimate the cloud fraction at high latitudes (Figs. 6d and 6g) compared to the observation (Fig. 6b)
and reanalysis (Fig. 6c). These biases are mitigated in the FD simulations (Figs. 6e and 6h), as the cloud fractions are limited
due to insufficient water vapor content at high latitudes. Despite more clouds being diagnosed at low levels over the eastern
subtropical ocean regions, the zonal mean cloud fraction profiles in the ALL simulations (Figs. 6f and 6i) are generally similar
to those from the FD simulations. In summary, the cloud fraction profiles have been improved from the LS to ALL simulations
due to the freeze-dry adjustment and the extra low clouds. However, the cloud fractions are still overestimated in high levels
over the subtropics, which could possibly explain the CRE biases over these regions.
In addition to the cloud fraction profiles, the geographic patterns of cloud amount, diagnosed from the random-maximum
overlap assumption (Sect. 2.2.4), are also compared with observations. For example, the annual mean spatial patterns of low





cloud amount from three different simulations (LS, FD and ALL) with linear RH cloud scheme, ISCCP-H data set, and the differences between them are shown in Fig. 7. It should be pointed out that the simulated cloud amounts are compared to the satellite retrievals directly in this study, because the cloud simulator (e.g., Bodas-Salcedo et al., 2011) has not been implemented

in Isca. This may bring in some inconsistencies, but it does provide a useful first look of the cloud amount diagnosis.

One evident feature of the low cloud amount in observations is that marine stratocumulus clouds dominate the areas off the west coast of continents (Fig. 7d), related to the subsidence branch of the Hadley Cell (Wood, 2012). The predominantly downward motion in these regions generally suppresses cloud formation in the middle and upper troposphere, but due to the abundance of water vapor near the ocean surface, clouds form at the top of convective boundary layers. However, these marine

low clouds are too far from the coasts in the LS simulation compared to the observations (Fig. 7a). Looking at the differences between LS simulation and observations (Fig. 7g), the low cloud amount is underestimated by about 20% off the west coast of Peru. In fact, these are well-known biases in CMIP5 models (Dolinar et al., 2014). Another problem of the LS simulation is the overproduction of low cloud amount in polar regions (Fig. 7g). For example, LS simulation produces more than 40% low cloud over Arctic region.

In contrast, the cloud fractions in the FD and ALL simulations are adjusted by the freeze-dry method (see Sect. 2.2.2), which is mainly designed to reduce the unrealistic cloud amount in polar regions. Thus there is a reduction of low cloud amount over high latitudes in these two simulations (Figs. 7h and 7i), although some positive biases still exist there. Compared with the LS simulation directly, the FD simulation can reduce the low cloud amount by more than 20% over polar regions (Fig. 7e), showing a better agreement with the observations. The ALL simulation can further diagnose the marine stratus clouds off the

west coast of continents through the predictor ELF, making the low clouds distribution closer to the observation (Fig. 7c). It is noted that pronounced changes occur off the west coasts of Peru, California and Namibia in the ALL simulation, where the cloud fraction increases over 20% (Fig. 7f) than the FD run. As shown in Table 3, the global mean low cloud amount decreases from 54.6% to 48.6% from the LS to ALL simulations with the linear RH scheme, which is closer to the observed value (28%). The changes of total cloud amount in these simulations (not shown here) are similar, and the global mean value decreases from

75.6% (the LS run) to 66.0% (the ALL run) for the linear RH scheme (Table 3).

The above analyses have shown that the freeze-dry method can improve the spatial patterns of annual mean cloud amount, with these changes being especially pronounced during winter time (as also noted by Vavrus and Waliser, 2008). Figure 8 illustrates the annual cycle of low and total cloud amounts over Arctic region from both linear RH and Sundqvist schemes. In the LS simulations, both the low and total cloud amounts are nearly at the same level throughout the year. However, a striking

feature in the FD simulations is that the cloudiness declines rapidly during boreal winter but remains almost unchanged in warm and moist summer, which in fact is more realistic compared to observations as pointed by Vavrus and Waliser (2008).

## 4.2 Cloud water path

The cloud water path (CWP) measures the total amount of cloud water within a column and is defined as the integral of cloud water content from surface ($p = p_s$) to TOA ($p = 0$) (Eq. 9.30 in Stensrud, 2007), and it can be expressed as follows if the



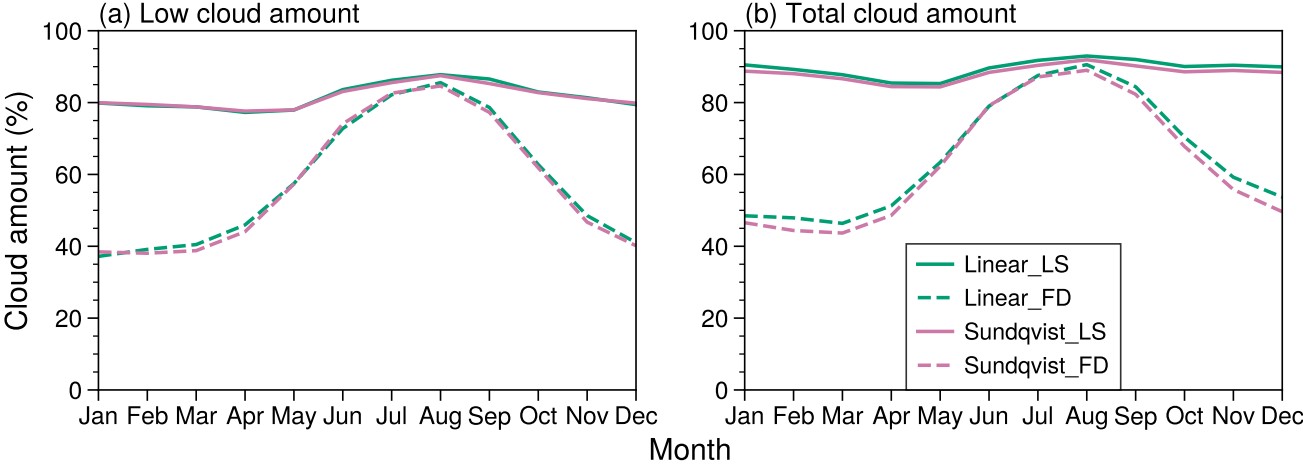

**Figure 8.** The seasonal cycle of (a) low and (b) total cloud amount (%) over the Arctic region (60°-90°N) from the LS (solid lines) and FD (dashed lines) simulations, where the freeze-dry adjustment method is applied in the FD simulations. The green and pink colors denote the experiments performed with the linear and Sundqvist cloud schemes respectively.

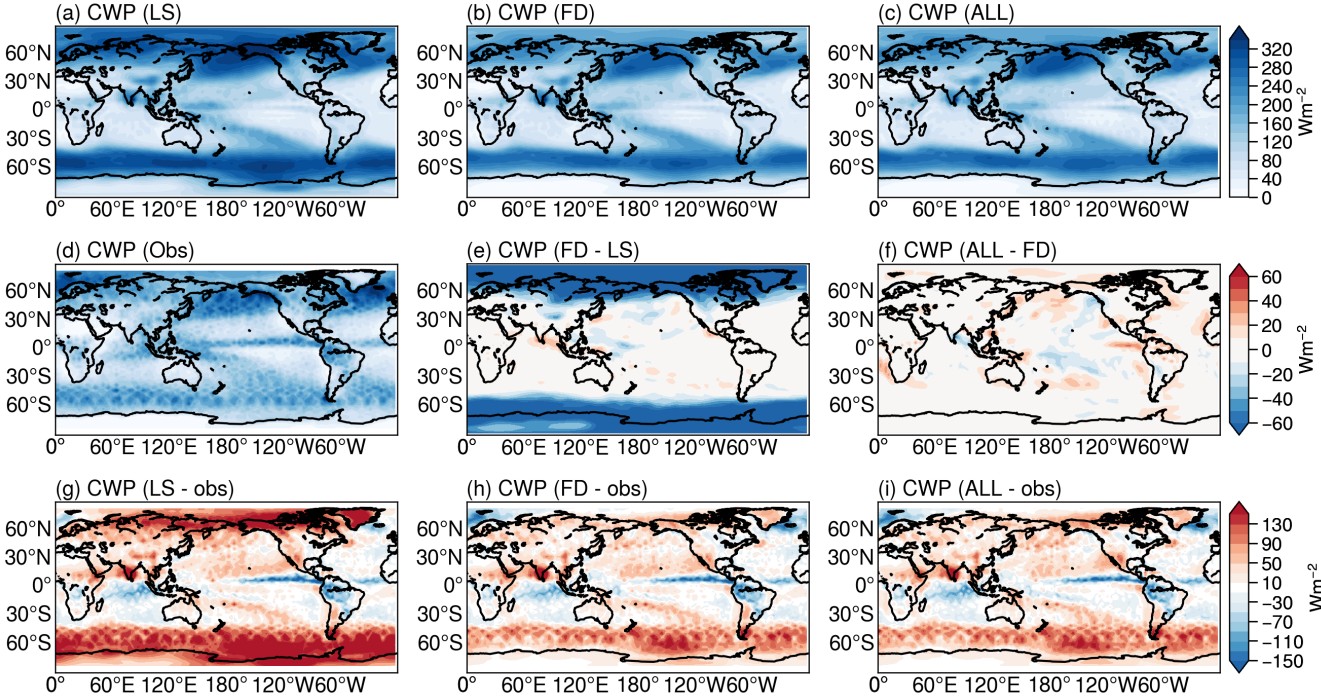

**Figure 9.** Same as Fig. 7, but for the spatial patterns of total cloud water path (CWP; $gm^{-2}$). The observed climatology of CWP is derived from the CloudSat data set.





hydrostatic equilibrium is assumed:

$$CWP = \int\limits_{p=0}^{p=p_s} C \cdot w_l \frac{dp}{g}, \tag{13}$$

where $w_l$ is the in-cloud liquid water mixing ratio specified in Equation (12), $C$ is the cloud fraction within a grid box, $g$ is the acceleration due to gravity and $p$ is the pressure. The global and annual mean CWP in the LS simulation from linear RH scheme is 147.0 gm$^{-2}$, which is larger than the observed global mean result (119.3 gm$^{-2}$, see Table 3). As displayed in Fig. 9,

one obvious bias in the spatial pattern in the LS simulation is the overestimation of CWP at high latitudes. For instance, these biases can be even more than 90 gm$^{-2}$ in the polar regions (Figs. 9a and 9g). Such an overestimation is also evident in cloud amount over the polar regions (e.g. Fig. 7g), suggesting that the adjustment of cloud fraction probably reduces the CWP biases there. Indeed, incorporating the freeze-dry method into the simulation produces a large change in the CWP spatial pattern, with a reduction over 60 gm$^{-2}$ over polar regions (Figs. 9b, 9e and 9h). The CWP biases off the west coast of continents are

reduced in the ALL simulation due to the increase of the low cloud fraction there. For example, the CWP over Peruvian and Californian coasts in the ALL simulation increases at least 20 gm$^{-2}$ when compared to the LS simulation (Fig. 9e).

### 4.3    Cloud radiative effect at TOA

The CRE is defined as the differences in TOA radiative fluxes between clear-sky and all-sky conditions (Ramanathan et al., 1989; Li et al., 2017, e.g.,). Specifically, the simulated LW CRE is derived from the difference between the outgoing longwave

radiation flux under clear-sky and all-sky conditions, and the SW CRE is computed from the difference in reflected SW flux under clear-sky and all-sky conditions. The net CRE is defined as the sum of LW and SW CREs.

### 4.3.1    Spatial patterns of cloud radiative effect

The global mean SW CRE from the LS simulation is -58.8 Wm$^{-2}$, which is much larger than the observed value of -45.8 Wm$^{-2}$ from CERES-EBAF (Table 3). Compared to the observed SW CRE (Fig. 10c), the LS simulation can reproduce the general

features of spatial patterns (Fig. 10a), although it fails to grasp some key features. For example, SW CRE is underestimated by over 30 Wm$^{-2}$ in eastern subtropical ocean basins off the west coast of Peru and over 15 Wm$^{-2}$ off the west coast of California (Fig. 10g), consistent with the insufficient low cloud amounts in these marine stratocumulus areas (Fig. 7g). These biases also exist in the FD simulation (Figs. 10b and 10h), as the freeze-dry method can only adjust the cloud amount over high latitudes. As shown in sections 4.1 and 4.2, the low cloud amount and CWP in these regions increase in the ALL simulation, which is

thus expected to improve the SW CRE biases. In fact, the differences between the ALL and FD simulations show that the SW CREs reduce by more than 10 Wm$^{-2}$ off the Californian, Peruvian and Namibian coasts (Fig. 10f). Consequently, the positive biases in SW CRE over eastern subtropical ocean regions are reduced, although some smaller positive biases still remain (Fig. 10i). The SW CRE biases from the LD and ALL simulations in the five marine stratocumulus clouds regions (defined in Fig. 5) are quantified in Fig. 11a. It is clear that these biases are reduced in all the locations, which is closely linked to the increase

of low cloud amount over these regions (Fig. 11b).



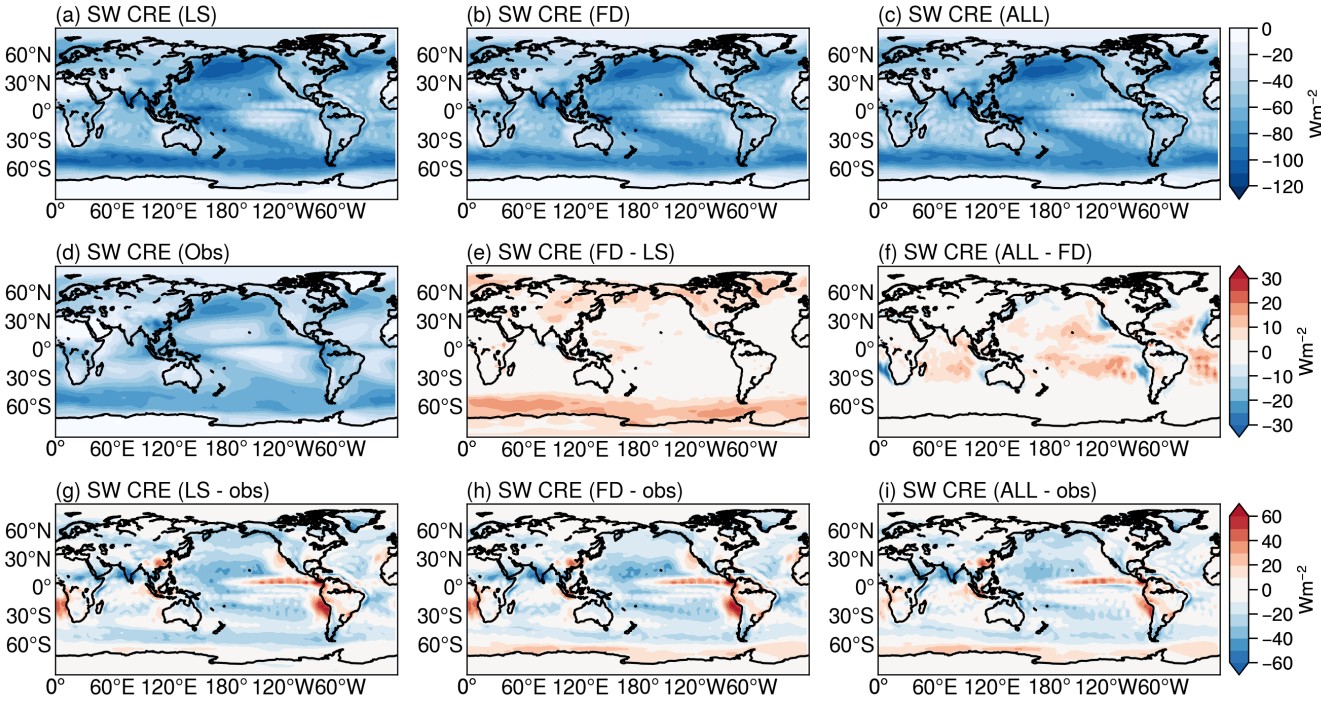

**Figure 10.** Same as Fig. 7, but for shortwave (SW) cloud radiative effect (CRE) (Wm$^{-2}$) at TOA. The observed SW CRE is from CERES-EBAF data set.

Another problem of the SW CRE in the LS simulation is that it is too negative in trade wind cumulus regions, Southern Ocean and northern Pacific Ocean (Fig. 10g), which is associated with the excessive clouds over these regions (Fig. 7g). The freeze-dry adjustment has reduced the cloud amount at high latitudes, making the SW CRE in the Southern Ocean less negative compared to the LS simulation (Figs. 10e and 10h). In the end, the SW CRE in the ALL simulation becomes more

realistic compared to observations, and the global mean value (-56.0 Wm$^{-2}$) is closer to the observed value (-45.8 Wm$^{-2}$) from CERES-EBAF (Table 3).

The LS simulation reproduces the general spatial pattern of the observed LW CRE (Figs. 12a and 12d). However, the radiative effect is too strong, especially in the polar regions and also over the maritime continent regions (Figs. 12a and 12d), which is also illustrated by the positive biases over these regions (Fig. 12g). The LW CRE in the LS simulation is overestimated by over

30 Wm$^{-2}$ in the Arctic and over 15 Wm$^{-2}$ in tropical regions. As discussed in previous sections, the cloud fraction, as well as the CWP in polar regions, decreases in the FD simulation compared to the LS run. Therefore, the LW CRE is improved over these regions (Figs. 12b and 12h), where the bias in polar region is reduced by more than 15 Wm$^{-2}$ (Fig. 12e). Nevertheless, there is still a small positive bias over the Arctic and tropical regions. Compared to the FD simulation, the changes in the ALL simulation has little effect on LW CRE (Fig. 12f). After these improvements, the spatial patterns of LW CRE in the FD and



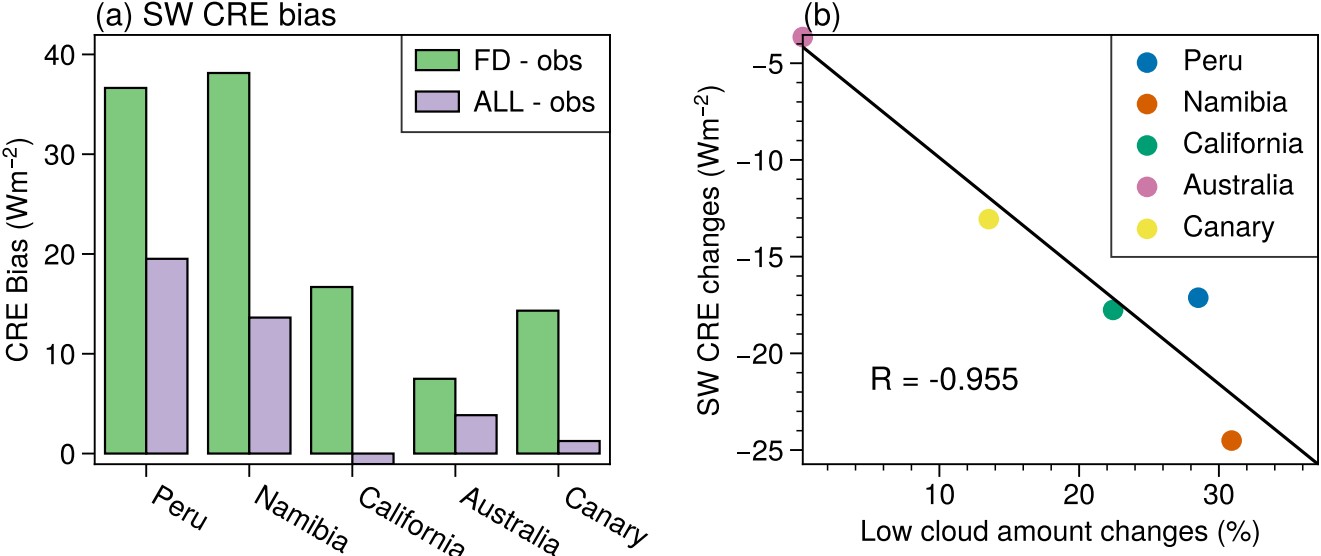

**Figure 11.** (a) The regional mean SW CRE biases from the FD and ALL simulations with with linear RH scheme in five different subtropical ocean regions off the west coast of continents, whose ranges are defined in the caption of Fig. 5. (b) The relationship of regional mean SW CRE and low cloud amount changes in the FD and ALL simulations, and the changes are calculated as their differences (i.e., ALL-FD).

ALL simulations become more similar to the observations, and the global mean CRE drops from 36.4 Wm$^{-2}$ to 31.0 Wm$^{-2}$, much closer to global mean result from observations (Table 3).

To further quantify the simulated LW CRE at TOA over the tropical ocean regions (30°S-30°N), following the method employed in Bony et al. (2004) and Bony (2005), we first define the upwelling and downwelling regimes based on the vertical pressure velocity at 500 hPa ($\omega_{500}$, Fig. 13a), and then evaluate the LW CRE over these regimes. $\omega_{500}$ is a measure of the

large-scale atmospheric circulation, where the regions with positive $\omega_{500}$ are associated with the subsidence movement, while those with negative $\omega_{500}$ are related to large-scale atmospheric ascent. The PDFs of $\omega_{500}$ from the ERA-Interim reanalysis and Isca simulations (LS, FD and ALL runs) are displayed in Fig. 13b. The PDFs of the Isca simulations generally follows the observations, albeit the Isca simulations have more weakly ascending regions and fewer weakly descending regions. The peak values of PDFs are located at 5-20 hPa/day, consistent with the results from Bony et al. (2004).

Figures 13c and 13d illustrate the high/low cloud amounts and LW CRE at the TOA over different dynamical regimes over tropical oceans, respectively. The observed cloud amount and LW CRE are from ISCCP-H and CERES-EBAF data sets respectively, both covering the period from 2005 to 2014 with the regimes being defined by the $\omega_{500}$ from ERA-Interim. The regimes with stronger convective activity, related to the magnitude of $\omega_{500}$ in ascending regions ($\omega_{500} < 0$), usually have a larger amount of high clouds and thus stronger LW CREs. All the LW CREs from the three simulations are close to the

observed values over the weak upwelling and subsidence regions. However, the LW CREs from the LS simulation deviate from the observations in strong ascending regions ($\omega_{500} < -20$ hPa day$^{-1}$). Furthermore, this discrepancy increases with the



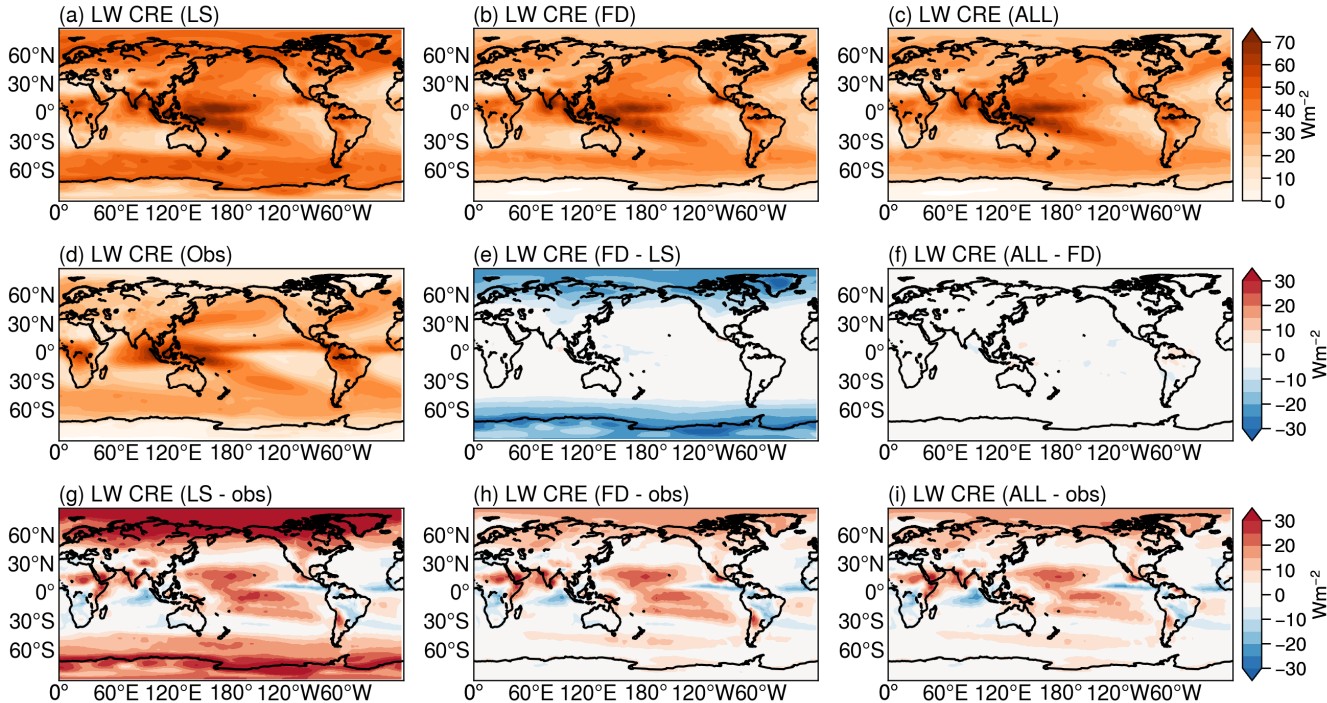

**Figure 12.** Same as Fig. 10, but for LW CRE (Wm$^{-2}$) at TOA.

magnitude of $\omega_{500}$ in ascending regions ($\omega_{500} < 0$). It is noted that the large biases of LW CRE over ascending regions is reduced slightly in the FD and ALL simulations, associated with the decrease of high clouds over those regimes (Fig. 13c). However, the positive biases still exist at the strong convection regions.

Finally, the spatial pattens of net CREs at the TOA are presented in Fig. 14, where we can see that the positive biases in the LS simulation mainly occur in the polar regions and subtropical eastern ocean regions. There are also small negative biases in subtropical and extratropical regions. The positive biases in net CRE in the LS simulation are related to the cloud amount biases in these regions, as we have seen in the SW and LW CRE fields. Clearly, the biases in polar regions are reduced greatly in the FD simulations (Figs. 14b, 14e and 14h) due to the freeze-dry method. Additionally, the positive biases off the west coasts of

continents in subtropics can be mitigated in the ALL simulation (Fig. 14i), making the spatial pattern closer to CERES-EBAF, although there are still slight positive biases in polar regions.

### 4.3.2   Zonal mean structure

To further study their latitudinal variations, the zonally averaged SW, LW and net CREs from Isca simulations, CMIP5 simulations, satellite observation and reanalysis data set are shown in Fig. 15. For the SW CRE (Fig. 15a), the general latitudinal

variations can be captured by all the Isca simulations, but the magnitude is larger than observations. The largest discrepancy in the LS simulations occurs in the mid-latitudes, especially in the Southern hemisphere, which is likely arising from the ex-



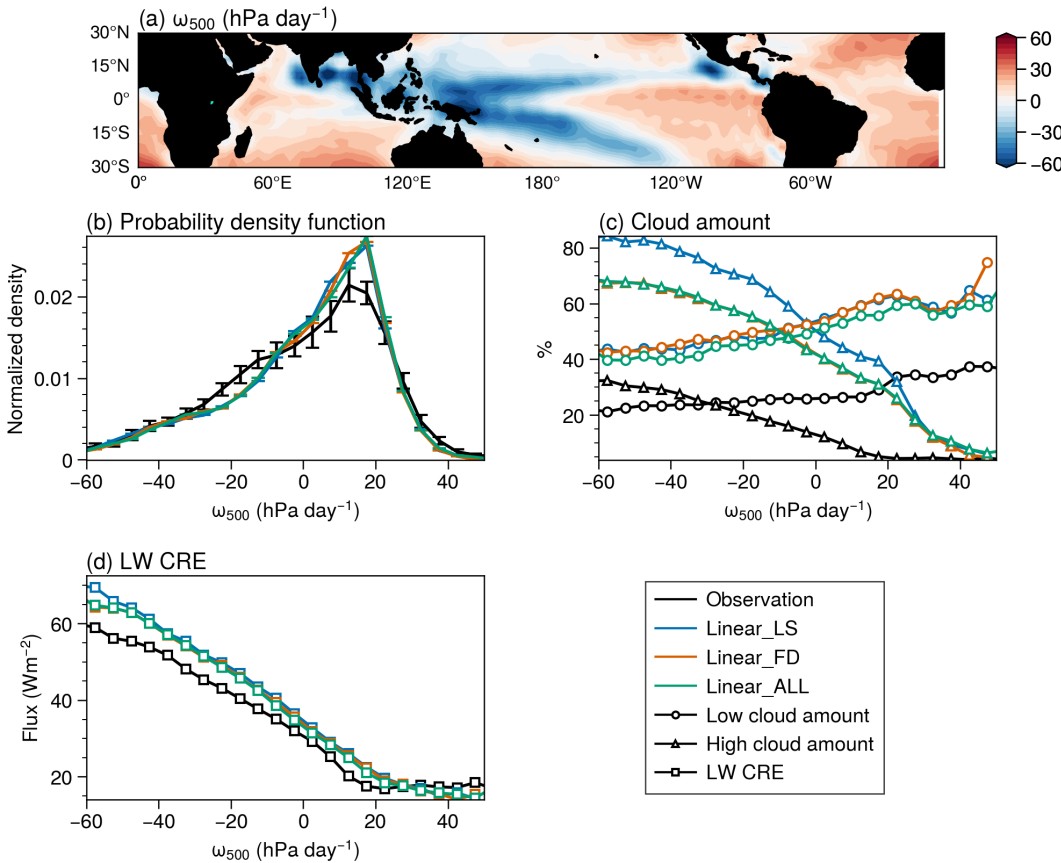

**Figure 13.** (a) The vertical pressure velocity field at 500 hPa ($\omega_{500}$) over tropical oceans between 30°S and 30°N from the linear_LS simulation. (b) The probability density functions (PDFs) of the 500 hPa large-scale vertical velocity ($\omega_{500}$) over the tropical ocean regions defined in (a), where the vertical bars indicate one standard deviation of the annual mean data. (c) The low, high cloud amounts and (d) the TOA LW CRE in different dynamical regimes binned by $\omega_{500}$. The 10-year (2005-2014) observed cloud amounts from ISCCP-H and the LW CRE from CERES-EBAF are binned by $\omega_{500}$ from ERA-Interim reanalysis data set (black lines). The results from the LS, FD and ALL simulations with the linear RH scheme are represented by blue, orange and green lines, respectively.



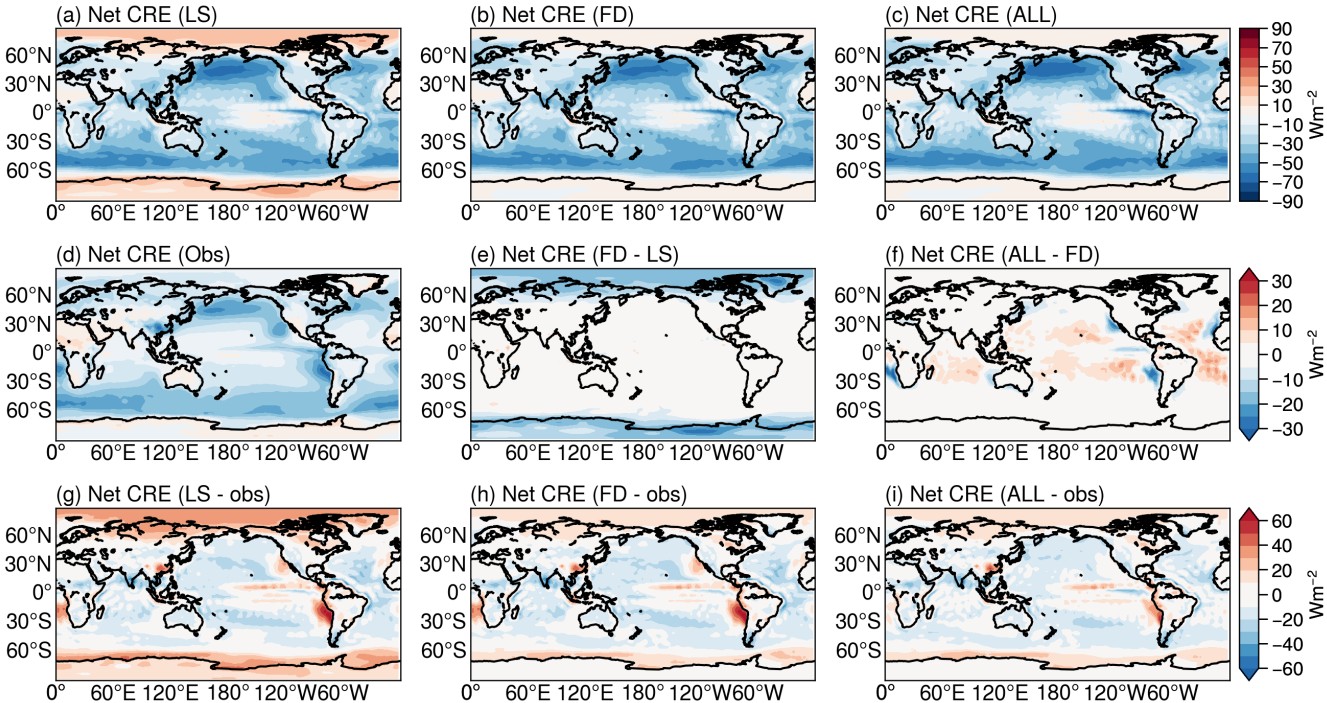

**Figure 14.** Same as Fig. 10, but for net CRE (Wm$^{-2}$) at TOA.

cessive cloud amount over these regions (Fig. 7g). The improvement of cloud amount biases in the FD and ALL simulations contributes to the improvement of SW CRE over the extratropics. However, the difference of zonal mean SW CREs between the FD and ALL simulations is slight, although the SW CRE biases over eastern subtropical ocean regions are reduced in the

ALL run (Fig. 10f). In addition, the remaining SW CRE biases, as well as the low cloud amount biases, in the ALL simulation over the subtropics and extratropics might be alleviated by an 'omega correction', namely a reduction of the low cloud fraction if subsidence is strong (Gordon, 1992, e.g.,), but the effects are mixed and we do not include that process in these results.

Regarding to the LW CREs (Fig. 15b), the LS simulations with both the linear RH and Sundqvist schemes agree well with observations at low latitudes, but show large discrepancies from observation from mid to high latitudes, which is consistent

with the large biases of cloud amount at high latitudes (Figs. 6d and 6g). It is striking that these biases can be largely reduced through the freeze-dry adjustment, as the LW CREs agree much better with the observation at high latitudes in the FD and ALL simulations. The remaining deviation from observation in Isca simulations over Arctic region is possibly associated with the simple sea ice setup in our model. Likewise, the disagreement between zonal mean net CREs at high latitudes between the LS run and the observations almost disappears in the FD and ALL runs (Fig. 15c).

In addition, compared to the zonal mean variation of the SW, LW and net CREs from CMIP5 models, the Isca simulations are generally located within the minimum and maximum range of the CMIP5 simulations at each latitude, except the LW



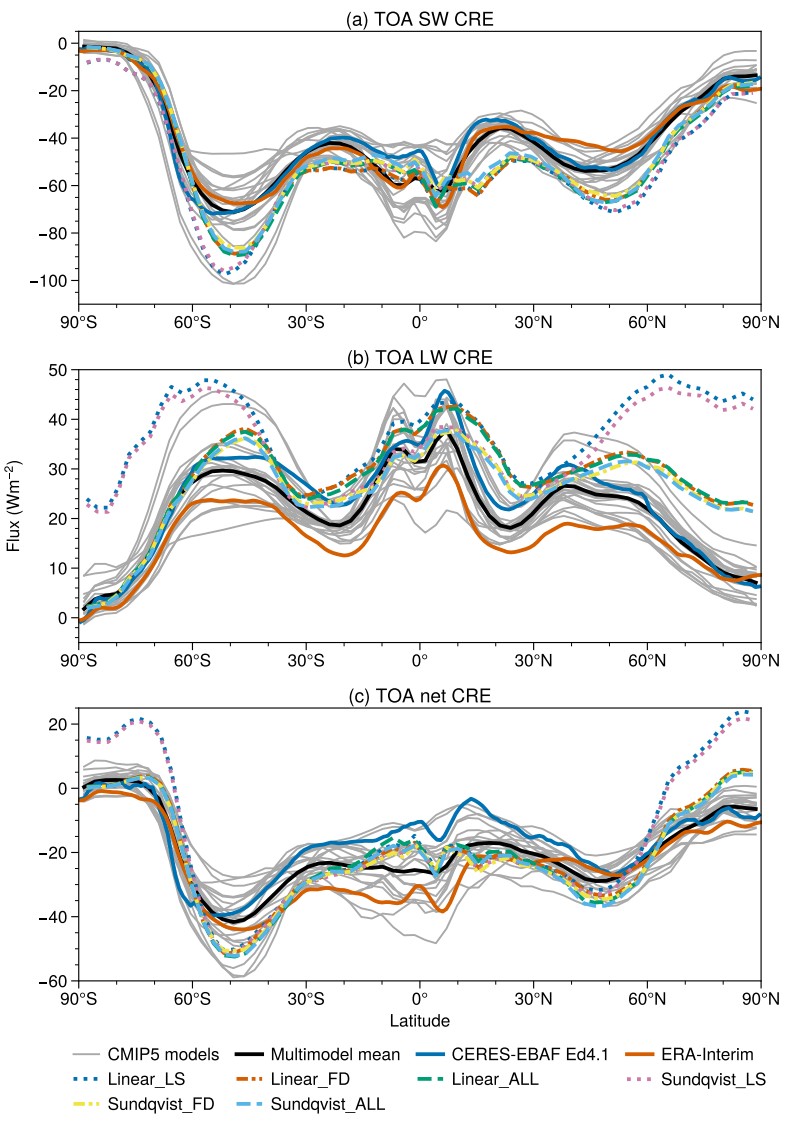

**Figure 15.** Zonally averaged distribution of the TOA (a) SW, (b) LW and (c) net CREs from CERES-EBAF Ed4.1 (blue solid line), ERA-Interim reanalysis (orange solid line) , CMIP5 models (thin gray solid lines for each model and black solid line for multimodel mean) and different Isca simulations (dashed/dotted color lines, listed in legend).



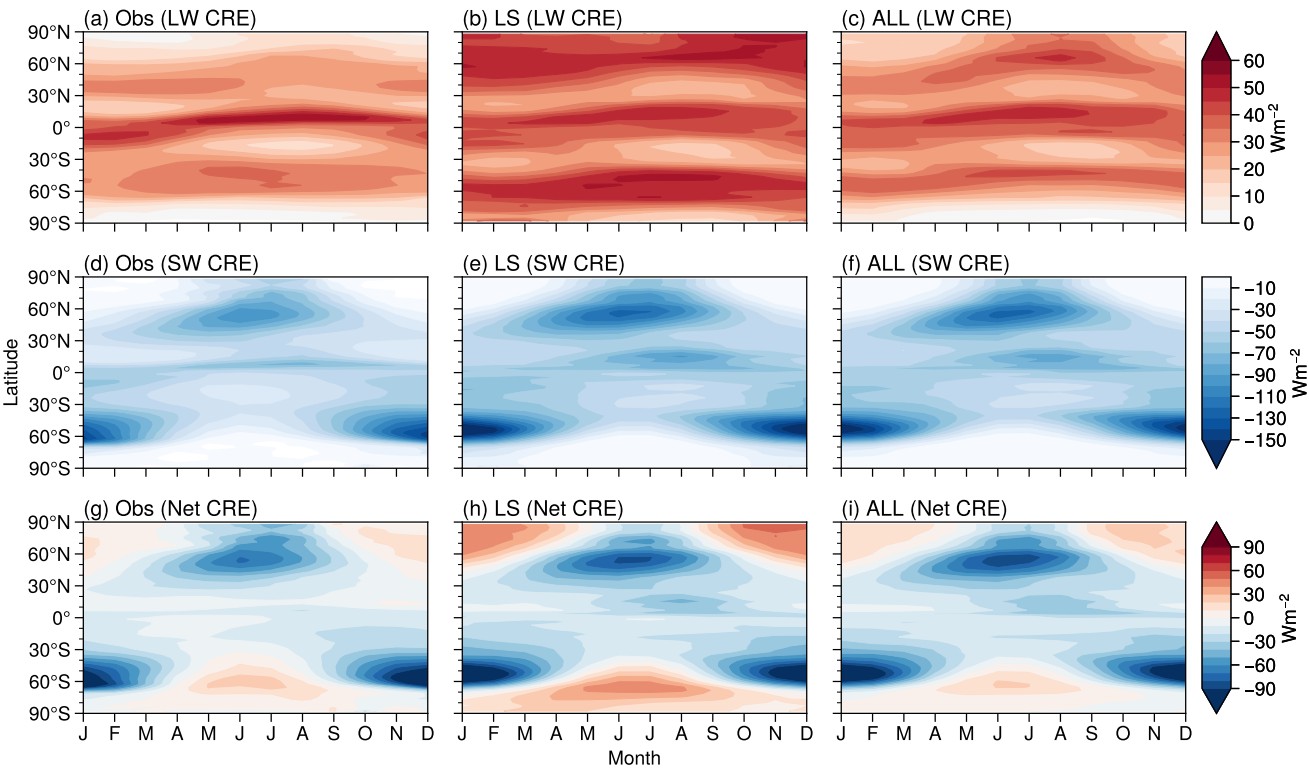

**Figure 16.** The zonal mean annual cycle of TOA LW (top), SW (middle) and net (bottom) CREs from observation (CERES-EBAF), LS and ALL simulations with the linear RH scheme in Isca.

and net CREs over high latitudes in the LS simulations (Figs. 15b and 15c). These biases are alleviated in the FD and ALL simulations, although there are still some discrepancies over Arctic regions.

### 4.3.3 Seasonal cycle

The zonal mean seasonal cycles of CREs from CERES-EBAF and Isca simulations (LS and ALL) with the linear RH scheme are displayed in Fig. 16. In the Arctic region, the observed LW CRE is weak during boreal winter and early spring, and has a maximum in summer (Fig. 16a). The simulated LW CRE tends to be overestimated throughout the year in the LS run (Fig. 16b), but the biases are alleviated by the freeze-dry adjustment (in the ALL run), particularly in winter (also see Fig. 8), leading to an overall improvement in the representation of the high-latitude seasonal cycle of the CRE. The existing problem for the

seasonal cycle of LW CRE is that the band in tropical region is too broad compared to the observations, which might relate to the too-broad high cloud pattern in tropical and subtropical regions (see Fig. 6f).

     The seasonal cycle of SW CRE in the LS simulation is good, except that it is too strong during boreal summer near $60°$N (Figs. 16d and 16e). This effect is slightly mitigated in the ALL simulation (16f) because of the improvement of cloud amount.



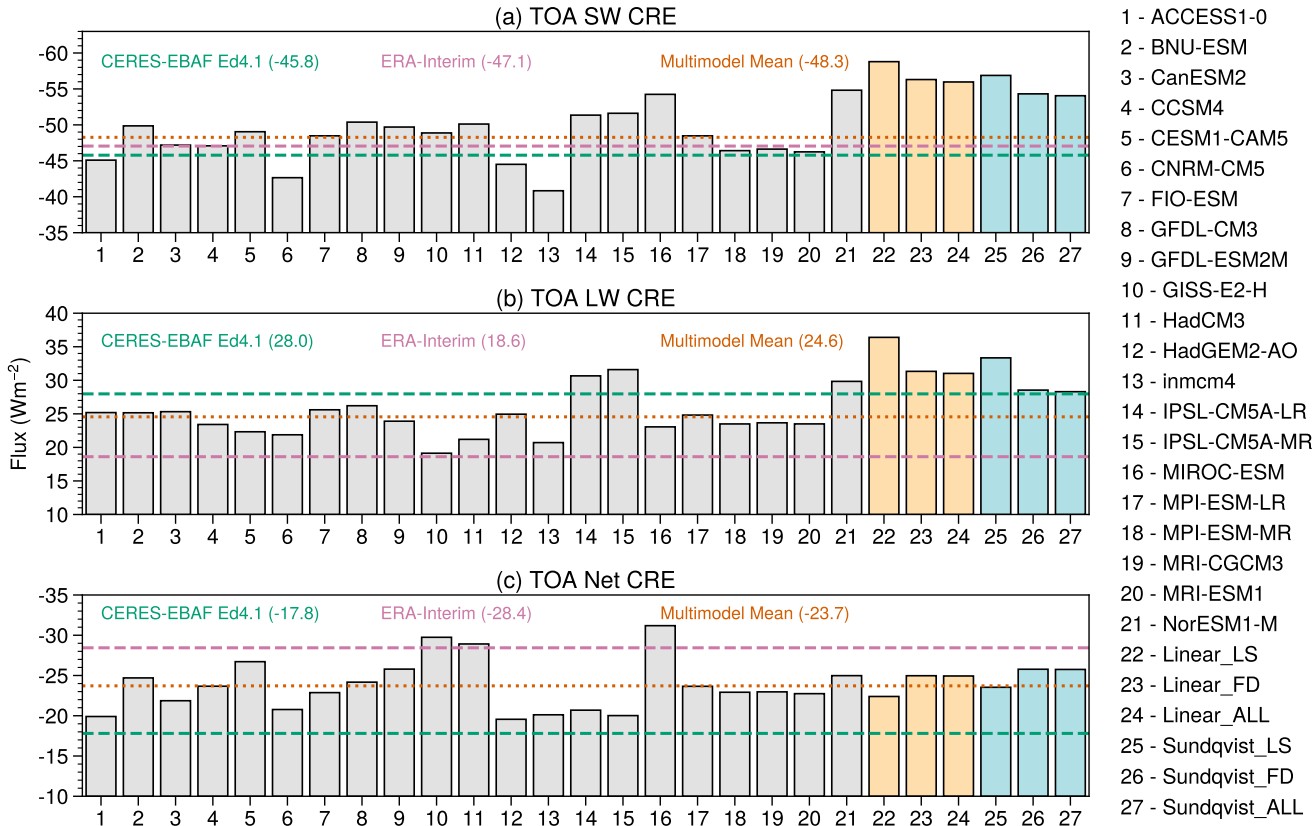

**Figure 17.** Globally averaged TOA (a) shortwave (SW), (b) longwave (LW) and (c) net cloud radiative effects (CREs, Wm$^{-2}$) from 21 CMIP5 models historical runs (1996-2005, gray bars) and Isca simulations with different setups (orange bars for linear scheme and cyan bars for Sundqvist scheme). The horizontal lines are annual and global mean CREs from CERES-EBAF (green dashed lines, covering from 2001 to 2018) and the multimodel ensemble mean results (orange dotted lines) of CMIP5 models, whose names are listed on right for reference.

Similar to the LW CRE, the positive biases of net CRE in winter over polar regions are also alleviated due to the improvement

of LW CRE in winter (Figs. 16h and 16i). In summary, the seasonal cycles of LW, SW and net CREs in simulations with freeze-dry and inversion-based adjustments compare well to observations (left and right columns of Fig. 16), indicating that the cloud scheme does reproduce a reasonably realistic seasonal cycle of CREs.

## 4.4 Comparison of CREs with CMIP5 models

To evaluate the simulated CREs further, Isca simulations are compared with CMIP5 models. Figure 17 shows the global mean

TOA SW, LW and net CREs from 21 CMIP5 models and Isca simulations with different cloud scheme setups. The observed SW CREs from CERES-EBAF and the multimodel mean of CMIP5 models are -45.8 Wm$^{-2}$ and -48.3 Wm$^{-2}$ respectively. While the multimodel mean SW CRE shows small difference from the observation, the spread among these CMIP5 models is





**Figure 18.** Taylor diagrams showing standard deviation (Wm$^{-2}$), root mean square error (RMSE; Wm$^{-2}$) and spatial pattern correlation for the observed and simulated (a) LW, (b) SW and (c) net CREs at TOA in CMIP5 models and Isca simulations (LS, FD and ALL). The statistics of these variables are calculated based on annual mean data, where the monthly data (1996-2005) from historical simulation is used for analysis of CMIP5 models. The observed field is as a reference and denoted by a black star. Contour of the standard deviation from observed field is shown by the black dashed line and contours of RMSE are displayed in gray with labels.





**Table 4.** Global and annual mean climatology from parameter sensitivity tests. In each test, only the parameter listed in the table header (see Table 1) is changed from the default values. The units for cloud effective radius ($r_e$) and in-cloud liquid water mixing ratio ($w_l$) are $\mu$m and g kg$^{-1}$ respectively.

|  | default | $r_{e\_liq} = 16$ | $r_{e\_liq} = 12$ | $r_{e\_ice} = 30$ | $w_{l0} = 0.15$ |
|---|---|---|---|---|---|
| TOA net SW flux (Wm$^{-2}$) | 229.6 | 234.2 | 227.1 | 231.9 | 236.0 |
| TOA net LW flux (Wm$^{-2}$) | 224.9 | 224.9 | 224.5 | 225.9 | 226.5 |
| TOA net flux (Wm$^{-2}$) | 4.7 | 9.3 | 2.5 | 6.1 | 9.5 |
| TOA SW CRE (Wm$^{-2}$) | -56.0 | -53.5 | -60.6 | -55.8 | -51.7 |
| TOA LW CRE (Wm$^{-2}$) | 31.0 | 31.1 | 31.2 | 30.3 | 29.6 |
| TOA net CRE (Wm$^{-2}$) | -24.9 | -22.4 | -29.4 | -25.5 | -22.1 |
| Cloud water path (gm$^{-2}$) | 130.4 | 130.7 | 130.6 | 131.7 | 108.7 |

large. Compared to the observation and CMIP5 models, the global mean SW CREs from the LS simulations with the linear RH and Sundqvist schemes are too strong, but they decrease in the FD and ALL simulations. With all components of our simple

cloud scheme (ALL simulation), the global mean values are -56.0 and -54.1 Wm$^{-2}$ for linear RH and Sundqvist schemes respectively, which are fairly close to the observed and multimodel mean values. The changes of LW CRE from the LS to ALL simulations are similar to SW CRE, where it drops from 36.4 to 31.0 Wm$^{-2}$ for linear RH scheme and decreases from 33.3 to 28.3 Wm$^{-2}$ for Sundqvist scheme, making the results from simple cloud scheme closer to observations. These changes are probably due to the decrease of cloud fraction and cloud liquid water path discussed in sections 4.1 and 4.2. The net CREs

from all the Isca simulations are in a range that are comparable to the CMIP5 models, which are close to the multimodel mean but still over 7 Wm$^{-2}$ larger than CERES-EBAF.

We can also use a Taylor diagram (Taylor, 2001) to compare Isca with other models, as this summarizes the standard deviation, pattern correlation and root mean square error (RMSE) in a single plot. Figure 18 shows the statistics of the observed and simulated LW, SW and net CREs from CMIP5 historical simulations (1996-2005) and Isca simulations. Compared to

CMIP5 models, the LS runs from both the linear and Sundqvist schemes display large RMSEs and low spatial correlations for LW CRE field (Fig. 18a), likely a consequence of too much clouds in polar regions. Similarly, the net CREs in the LS runs also show larger RMSEs and standard deviations than most CMIP5 models (Fig. 18c). The FD and ALL simulations improve matters: going from the LS to FD simulations the RMSE decreases from 12.4 to 8.9 Wm$^{-2}$ and from 19.8 to 14.1 Wm$^{-2}$ for LW and net CREs respectively. For the SW CREs (Fig. 18b), compared to the LS runs, the RMSEs in the ALL runs have

decreased slightly in both linear and Sundqvist schemes. By these metrics, the simple cloud scheme is evidently achieving a level of performance comparable to a number of CMIP5 models.





### 4.5 Parameter sensitivity of the scheme

Thus far we have largely selected the various parameter values using observations. In order to test the sensitivity to these choices, a small number of simulations with different parameters are conducted. The simulations analyzed here are run for ten years and the last five years are used for analysis.

The parameters used in the simulation are the default values introduced previously; that is, the effective radii for liquid and ice clouds particles are $r_{e\_liq} = 14$ and $r_{e\_ice} = 25$ $\mu$m respectively, and the maximum in-cloud liquid water content one grid box can reach is $w_{l0} = 0.18$ g kg$^{-1}$. As displayed in Table 4, changing the value of the effective radius for liquid clouds has little impact on LW flux and CRE at TOA, but has large impact on those fields associated with the SW flux. For instance, the net SW flux at TOA has reduced (increased) by about 2.5 Wm$^{-2}$ (4.6 Wm$^{-2}$) when the effective radius of liquid cloud decreases (increases) from 14 $\mu$m to 12 $\mu$m (16 $\mu$m), which can be explained by the relationship between effective radius ($r_e$) and shortwave optical depth ($\tau$) of clouds, $\tau = 3\text{LWP}/(2\rho_w r_e)$, where LWP is the liquid water path of clouds and $\rho_w$ is the density of water (Morcrette and Fouquart, 1986). Specifically, if the liquid water path remains unchanged, then $\tau$ increases (decreases) with the decrease (increase) of $r_e$ (e.g., Slingo and Schrecker, 1982), implying that the reflected SW flux would increase (decrease), the net SW flux at TOA would decrease (increase) and the SW CRE would get more (less) negative.

As introduced in Sect. 2, ice clouds are treated as liquid clouds, except that the effective radius is different. Thus the increase of effective radius of ice clouds has similar effect of the increase of effective radius of liquid clouds, resulting in the increase of net SW flux and less negative SW CRE at the TOA (see the fourth column in Table 4). However, what is different is that tuning the effective radius of ice clouds can also have influence on the LW related flux and CRE, as the ice clouds are usually located at high levels.

In addition, when changing the maximum value of in-cloud water mixing ratio that a grid box can reach (i.e., $w_{l0}$) from 0.18 g kg$^{-1}$ to 0.15 g kg$^{-1}$, the global mean LWP decreases over 20 gm$^{-2}$ (last column in Table 4). In this case, if the $r_e$ is unchanged, then $\tau$ would decrease in response to the decrease of LWP. Hence, the atmosphere becomes less opaque, which has an opposite effect of reducing $r_e$ and thus the net SW flux at TOA increases and the SW CRE becomes less negative. Therefore, these parameters can be used to adjust the radiative properties associated with SW. We should notice that tuning $w_l$ can also have effect on LW radiative fluxes.

As mentioned in Sect. 2, the coefficient $a$ in the linear scheme is related to the critical relative humidity. Therefore, the parameter $a_s$ in Equation (2), which is related to the critical relative humidity at lower levels, can be used to tune the SW CRE. Changing the parameter $a_t$ in Equation (2), which determines the coefficient profile (and the critical relative humidity) at high levels, has a influence on the LW CRE, but also on the SW CRE (results not shown here). In general, all the parameters associated with the critical relative humidity ($a_s$ or $a_t$), effective radius and cloud liquid water content can be used to tune the SW and LW CREs, and users can determine which one to use based on their research purpose.





## 5 Discussion and conclusions

We have presented and explored a simple diagnostic cloud scheme, SimCloud, with two general aims. First, we hope to provide
a scheme whose dependence of various parameters and processes is transparent, so that it can be used to explore and understand
the cloud distribution and its possible changes, as well as possible differences among models. Second, we hope to provide a
cloud scheme at a modest level of complexity that can be used in GCMs in a broad range of situations where some basic
representation of cloud cover would be useful. For example, prior to the implementation of SimCloud, Isca did not have a
cloud scheme. In this study we have improved the radiative properties of Isca by introducing the cloud scheme.

This simple diagnostic scheme is not meant as a replacement for more complicated schemes that are based on microphysical
properties and/or explicitly on liquid and solid phases of the condensate. Rather, it is intended to be used in models which
may require a level of complexity commensurate with other parameterizations, and/or in situations where particular processes
are to be investigated. Cloud schemes in many comprehensive GCMs have become very complicated, and differ considerably
in detail from each other, and we think there is value in having a simpler scheme, but one that also has a number of realistic
features and that captures the observed cloud climatology with some verisimilitude.

In SimCloud scheme, the cloud fraction, the effective radius of the cloud droplet and in-cloud water mixing ratio are param-
eterized. The cloud fraction parameterization includes two kinds of clouds: large-scale clouds and marine low clouds, with the
addition of a 'freeze-dry' adjustment. The effective radius of cloud droplets is calculated as a weighted mean of liquid cloud
droplet and ice cloud crystal, with the weight specified by the liquid cloud fraction, which is defined as a linear function of
temperature. The in-cloud liquid water content is also determined as a function of temperature, where the temperature threshold
is deduced from the observed liquid and ice cloud top temperature. The equations and parameters used in this cloud scheme
are summarized in Table 1. The parameters themselves are user-configurable; here we merely present those that we have found
to be useful.

The large-scale clouds are diagnosed from relative humidity and there are two options for the scheme: The first is a linear
relationship of relative humidity, where the coefficient profile is derived from hourly reanalysis data sets; the second one has
the same form as the Sundqvist scheme. The code is quite flexible and other choices could readily be implemented. Simulations
with large-scale clouds only (LS simulation) show that this method does capture the basic features of spatial patterns of cloud
fraction and CREs at TOA. Using the linear relation connecting cloud fraction to relative humidity gives similar results to those
from the Sundqvist scheme (which uses a square root dependency). However, both relative-humidity schemes were found to
have two deficiencies when used by themselves. The first is that the cloud cover is generally too high in the high latitudes,
especially over polar regions, which in turn leads to an overestimated CRE over these regions. These biases can be found not
only in annual mean spatial patterns, but also in the seasonal cycles. The second issue is the underestimation of cloud fractions,
and hence the SW and net CREs, in the marine stratocumulus regions off the west coast of continents; this has also been a long
standing problem in CMIP3 and CMIP5 models (Dolinar et al., 2014).

In order to mitigate the biases of extra clouds over polar regions, a modified freeze-dry method from Vavrus and Waliser
(2008) may be used, in which the cloud fractions over high latitudes are adjusted by a function of specific humidity. With this





method, the seasonal cycle of cloud fraction over Arctic was found to be well simulated and the cloud fraction reduced to fairly realistic levels largely during boreal winter. The improvement of the cloud fraction over high latitudes also decreases the CRE biases, contributing to the improvement of the seasonal cycle of LW CREs. We should note that in this adjustment the specific

humidity threshold are derived according to current climate, but whether the threshold holds under global warming still needs further investigation. To alleviate the problem of the low cloud biases, a diagnostic low cloud scheme based on a proxy called ELF (estimated low-cloud fraction) from Park and Shin (2019) was implemented, as the ELF shows a linear relationship with low cloud fraction in the reanalysis data set. The simulation with both large-scale clouds and low clouds (the ALL simulation) reduced the SW and net CRE biases off the west continental regions over subtropics, by increasing both the cloud fraction and

the cloud water path.

In summary, many of the basic features of observed cloud fraction and cloud radiative properties are captured by the cloud scheme. Many metrics of the simulation results, incorporating both spatial and temporal variability, are comparable to those of a number of CMIP5 models that use more complicated schemes, suggesting that the scheme might be used to study problems related to cloud feedback and cloud-circulation coupling. In addition, the scheme is relative flexible and many aspects are

optional or user-configurable, so the users can decide which one to use as per their own research interests or purposes. For example, if the users do not focus on the polar regions, they can omit the freeze-dry adjustment.

Certainly, the scheme has deficiencies. For example, the short-wave CREs are still a little weak off the west coast of continents and too strong over the extratropics compared to observations. The global mean CREs, including the LW and SW CREs, are too strong, and the TOA radiation imbalance is rather large compared to the observations, which perhaps could be solved

by further improvements of the cloud scheme, such as the microphysical process.

Finally, at a more general level, the diagnostic scheme we have presented does not vary with model resolution and so is not 'scale aware'. Whereas this may be perfectly appropriate at low and modest resolutions, it would fail as the model resolution increases, for the grid-box distribution of relative humidity varies according to the resolution, and so the functional dependence of cloud fraction should properly be a function of grid size. This drawback is not, however, unique to our scheme. It could

be overcome at an empirical level by re-tuning the coefficients as resolution changes. We have not found this to be an issue in practice at the resolutions we have used. If desired, it may be possible to address this using a more sophisticated treatment of the distribution properties of humidity, with the width of the moisture distribution, and hence the critical value of relative humidity, then becoming a function of grid size and/or being dynamically determined. Separate schemes to take into account the low-level inversion and polar effects would then ideally not be needed. It would also be of interest to further simplify the

scheme so that it could, for example, be coupled to simpler (e.g., semi-gray) radiation schemes with less complicated treatments of scattering and absorption and which might be more appropriate for very different climate regimes. These are topics for future work.

*Code and data availability.* The code of SimCloud scheme can be found at https://github.com/lqxyz/Isca/tree/simple_clouds, and it will be mergred with Isca master repository (https://www.github.com/ExeClim/Isca) in the future. Please refer to the Supplement for a brief in-



troduction of the code structure. The Isca model outputs produced for this study are available on Zenodo: https://doi.org/10.5281/zenodo. 3831988. An archive of the scripts used to process data and generate figures/tables is available at https://doi.org/10.5281/zenodo.4319639 and the updates can be found at https://github.com/lqxyz/cloud_scheme_manuscript_figs. The reanalysis data sets used in this study are ERA-interim (https://www.ecmwf.int/en/forecasts/datasets/archive-datasets/reanalysis-datasets/era-interim) and ERA5 (https://cds.climate. copernicus.eu/cdsapp#!/home). The observed cloud fraction products used for analysis include ISCCP-H series (https://www.ncdc.noaa.gov/

cdr/atmospheric/cloud-properties-isccp), CALIPSO-GOCCP (https://climserv.ipsl.polytechnique.fr/cfmip-obs/Calipso_goccp.html) and Cloud-Sat (http://www.cloudsat.cira.colostate.edu/data-products/level-2b/2b-cwc-ro). The observed cloud radiative effect products are from CERES-EBAF (https://ceres.larc.nasa.gov/compare_products.php). The CMIP5 ouputs are obtained from the Centre for Environmental Data Analysis (https://www.ceda.ac.uk).

*Author contributions.* QL implemented the cloud scheme, performed the simulations and analyzed the results under the supervision of MC

and GKV. PM and SIT provided technical help in coding the cloud scheme. All authors contributed to the results analysis and the editing of the manuscript.

*Competing interests.* The authors declare that they have no conflict of interest.

*Acknowledgements.* We gratefully thank Paulo Ceppi, Mark Webb and other members of the phase II of SPOOKIE project for providing the in-cloud liquid water mixing ratio formula in Equation (12). QL is supported by the scholarship from University of Exeter and China

Scholarship Council (No. 201706210070). MC acknowledges the support from NE/N018486/1, PM is supported by NE/T003863/1, SIT is supported by the UK-China Research and Innovation Partnership Fund, through the Met Office Climate Science for Service Partnership (CSSP) China, as part of the Newton Fund, and GKV is supported by the Leverhulme Trust.



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
