# Peer review of "SimCloud version 1.0: a simple diagnostic cloud scheme for idealized climate models"

_Geoscientific Model Development, 2020_

## Short Comment (SC1) · 21 Dec 2020

Dear authors,

in my role as Executive editor of GMD, I would like to bring to your attention our Editorial version 1.2:

https://www.geosci-model-dev.net/12/2215/2019/

This highlights some requirements of papers published in GMD, which is also available on the GMD website in the 'Manuscript Types' section: http://www.geoscientific-model-development.net/submission/manuscript_types.html

In particular, please note that for your paper, the following requirement has not been

met in the Discussions paper:

- Code must be published on a persistent public archive with a unique identifier for the exact model version described in the paper or uploaded to the supplement, unless this is impossible for reasons beyond the control of authors. All papers must include a section, at the end of the paper, entitled "Code availability". Here, either instructions for obtaining the code, or the reasons why the code is not available should be clearly stated. It is preferred for the code to be uploaded as a supplement or to be made available at a data repository with an associated DOI (digital object identifier) for the exact model version described in the paper. Alternatively, for established models, there may be an existing means of accessing the code through a particular system. In this case, there must exist a means of permanently accessing the precise model version described in the paper. In some cases, authors may prefer to put models on their own website, or to act as a point of contact for obtaining the code. Given the impermanence of websites and email addresses, this is not encouraged, and authors should consider improving the availability with a more permanent arrangement. Making code available through personal websites or via email contact to the authors is not sufficient. After the paper is accepted the model archive should be updated to include a link to the GMD paper.

As GitHub is not a persistent archive, please provide a persistent release for the exact source code version used for the publication in this paper. As explained in https://www.geoscientific-model-development.net/about/manuscript_types.html the preferred reference to this release is through the use of a DOI which then can be cited in the paper. For projects in GitHub a DOI for a released code version can easily be created using Zenodo, see https://guides.github.com/activities/citable-code/ for details.

Yours, Astrid Kerkweg

---

## Short Comment (SC2) · 21 Dec 2020

Dear editor,

Many thanks for your comments about the code archive problem. We are sorry that we just published the codes on Github (without a persistent DOI) during the initial submission. Currently following your suggestions, a release of SimCloud scheme codes has been archived at Zenodo: https://doi.org/10.5281/zenodo.4382536, and we hope this could meet the requirements of GMD. Of course, this DOI will be added to the manuscript during the revision process.

In addition, the scripts used in the manuscript to analyze data and generate figures and tables have already been archived at Zenodo:https://doi.org/10.5281/zenodo.4319639,

which have also been included as part of the 'Assets'.

Best regards,

Qun Liu
* * *
**[GMDD](GMDD)**

---

## Referee Comment (RC1) · Anonymous Referee #1 · 7 Jan 2021

**1. General Comments**

This study proposes a simple diagnostic cloud scheme that could readily be adapted to multiple GCMs. The authors demonstrate that the cloud scheme can capture many of the basic features of observed cloud fraction and cloud radiative effects using an idealized GCM, Isca. Skill scores of the simulated results are comparable to many of the CMIP5 models, which is impressive. The proposal of the simple cloud scheme is useful, because the cloud scheme potentially helps to understand the inter-model difference in climate simulated by multiple GCMs, which has been a serious issue in climate science. The manuscript is well written. The description of the cloud scheme is sufficiently complete to allow reproduction by fellow scientists, although there is room for improvement. I recommend accepting the manuscript for publication after minor

revisions.

2. Specific comments

P1L20 "(e.g., Zelinka et al. 2017;...)" Maybe you could cite a study by Wild et al. (2019) who recently assessed cloud radiative effects. Wild et al. (2019), Climate Dynamics, 52, 4787-4812.

P2L27 "our best estimates are between..." Maybe you could mention the results of CMIP6, which is described by Zelinka et al. (2020). Zelinka et al. (2020), GRL, 47, e2019GL085782.

P5L122 "grid-mean relative humidity" Readers may be interested in the details of the relative humidity definition. Is it defined with respect to the liquid saturation only, or is it defined with respect to the ice saturation at temperatures below the freezing point? In addition, I suggest that the authors briefly describe how precipitation is calculated in the Isca GCM, because it may be relevant to the relative humidity distribution. I assume that the precipitation is diagnosed to remove super-saturation. Is the super-saturation with respect to liquid or ice?

P7L150 "0.95 at the surface, 0.85 at 700 hPa, and 0.99 at 200 hPa" Readers may be interested in how these values are determined.

P7L154 "a and b are determined from the least squares fitting" Are the parameters a and b dependent on pressure, as shown in Fig 2(a), or are they assumed constant globally?

P7L164 "biases in the cloud fraction and cloud radiative effect" Maybe you could also state that the biases are positive, namely overestimation, because you discuss reasons for the overestimation and how to reduce clouds in the following sentences.

P7L166 "there is little subgrid-scale heterogeneity of relative humidity" How does the subgrid-scale heterogeneity cause the overestimation of cloud fraction in the model simulation? It would be helpful to readers if you could describe the mechanism in more

detail.

P7L167 "The small quantity of condensation nuclei" How does the quantity of condensation nuclei cause the overestimation of cloud fraction in the model simulation? It would be helpful to readers if you could describe the mechanism in more detail.

P7L174 "only polar regions will be affected" It appears to me that the freeze-dry adjustment affects clouds in low latitudes as well, because cloud fraction is smaller in Fig 6(e) than in Fig 6(d) between 30S and 30N, at around 100 hPa level.

P9 Figure 4 caption "The thick solid and dashed black lines are specific humidity profiles" Maybe you could add that the profiles are from the Isca simulation, and whether they are annual averages or seasonal averages.

P12L245 "the liquid cloud fraction varies with temperature" Maybe you could reword the term "liquid cloud fraction", because "cloud fraction" has been used in the manuscript with a different meaning, namely, the areal fraction of a grid box that is covered with cloud.

P12L245 "which only has an influence on the effective radius" Readers may be interested in whether the "liquid cloud fraction" affects precipitation efficiency.

P13L266 "$3 \times 10^{-4}$ gkg$^{-1}$ at 220 K and wl0 = 0.18 gkg$^{-1}$" Readers may be interested in how these values are determined.

P14L283 "The sea ice data ... is averaged over all years and months" Readers may be interested in why you specified annual mean, not the monthly mean, of the sea ice distribution.

P21L403 "over the maritime continent regions" It appears to me that the positive bias of the LW CRE is pronounced over the subtropical oceans located east of the maritime continent, as shown in Fig.12g.

P31L525 "all the parameters associated with the critical relative humidity (a_s or a_t)"

It would be helpful to readers if you could suggest plausible range of the parameters (a_s and a_t), within which we are allowed to perturb the values.

3. Technical corrections

P1L4 "inversion strength" inversion height?

P1L4 "a simple function of relative humidity" specific humidity?

P6L148 "Cs=1-sqrt()" The definition of the Cs could be written as max(0, 1-sqrt()).

P19 Figure9, units shown near the color scales g mˆ-2 ?

P20L376 "Fig.9e" Fig.9ef?

P20L384 "Fig.10c" Fig.10d?

P20L393 "LD" FD?

P22L418 "Isca simulations have more weakly ascending regions and fewer weakly descending regions" Fewer weakly ascending regions and more weakly descending regions?

P35L619 "Bony, S." Bony, S. and J.-L. Dufresne?

---

## Referee Comment (RC2) · Anonymous Referee #2 · 18 Jan 2021

General comments:

This manuscript documents a simple diagnostic cloud scheme, SimCloud, which aims to understand the dependence of cloud fraction on different diagnostic schemes and tuning parameters. The SimCloud can switch various cloud schemes within a single model framework, facilitating the evaluation of the inter-model spread in cloud biases. The authors implement the SimCloud into an idealized climate model, Isca, and perform some sensitivity experiments to show how to evaluate cloud biases in the model. Although the schemes used in the SimCloud are very simple and not new, the tool will be of importance to understand the source of uncertainties in cloud and radiation fields among multiple models for future study. The study is generally well conducted and the methods used are appropriate. I feel that this work fits within the scope of the

[Figure]

Geoscientific Model Development. I recommend publishing this manuscript following minor revisions because the authors need to provide some additional discussion, and have several minor presentation issues to address detailed below.

Specific comments:

Abstract: Although the abstract is supported by data and clearly reviewed the main findings of the present study, it would be helpful to readers if the authors add a brief description of the purposes of the SimCloud (e.g., Lines 529-534).

Line 34: "The simulation of low clouds nevertheless remains problematic in GCMs": This paragraph mainly describes the bias in cloud feedback, but this paper particularly focuses on the climatological cloud biases under the present-day simulations. A brief discussion about systematic biases among GCMs (e.g., too bright low-cloud bias; Nam et al., 2012) would be beneficial to the readers.

Lines 83 and/or 102: More detailed description of the "freeze-dry" method should be provided. Or, please consider simply adding "(discussed in Sect. 2.2.2)" here.

Line 107-108: Does the cloud scheme predict cloud droplet number concentration (i.e., two-moment scheme)? Please clarify how the model represents the Twomey effect (cloud albedo effect). Also, please consider adding a description of the cloud lifetime effect.

Line 149-150: This may need references.

Line 265-266 and Eq. (12): This also may need reference(s).

Line 283: Just to check, "but seasonally varying" means that the authors used monthly SST data, right?

Sect. 3: How does the SimCloud/Isca model treat aerosol chemistry and aerosol-cloud interactions? Adding a brief description in this section would be helpful for readers.

Line 304 and Table 3: I assume that the "cloud water path" is the sum of liquid and ice.

Please specify the definition here.

Table 3 (and/or Fig. 7): Although the authors used cloud amount data from ISCCP, isn't it better to use CALIPSO-GOCCP data here?

Lines 320 and 333-335: In my experiences, cloud fraction emulated from satellite instrument simulator (lidar simulator) is fewer by approximately 5–10% than the model native output. This has also been reported in previous studies (e.g., Cesana and Chepfer, 2013). So the Isca simulations are actually close to the CALIPSO-GOCCP observations if using a simulator. Please consider adding this note here. Cesana, G., and H. Chepfer, 2013: Evaluation of the cloud thermodynamic phase in a climate model using CALIPSO-GOCCP. J. Geophys. Res. -Atmos., 118, 7922–7937.

Figure 9: Please modify the unit in the figure (Wm-2 => gm-2).

Line 371-373: I think that the overestimate of CWP can be caused by neglecting the ice- and mixed-phase microphysics scheme as well (e.g., Bergeron-Findeisen process).

Line 400-401 "is closer to the observed value": The net CRE bias is more than 10 W m-2, so this sentence is not appropriate. This result means that more fundamental errors can also exist in microphysical properties (e.g., effective radius) and/or other processes in the model in addition to the macrophysical properties (cloud fraction).

Line 578-580: In addition to the cloud scheme, recent studies showed that treatment of precipitation is also a very important component in GCMs. For example, some GCMs that incorporate the prognostic type of precipitation scheme (e.g., CAM, MIROC, etc.) have improved some systematic biases in the magnitude of aerosol-cloud interactions and rain formation processes with more realistic cloud and radiation fields. A brief discussion of the importance of process-based model developments against the simple model approach would be beneficial.
* * *
[Figure]

2020.

Interactive
comment

---

## Author Comment (AC1) · 19 Jan 2021

Many thanks for your insightful comments, which will definitely help us improve the manuscript. We will address your concerns during the revision and respond in full later.

---

## Author Comment (AC2) · 19 Jan 2021

Many thanks for your insightful comments, which will definitely help us improve the manuscript. We will address your concerns during the revision and respond in full later.

---

## Author Response (AR1)

**Response to Editor and Reviewers' Comments on**
**GMD-2020-402**

Qun Liu ⓘ, Matthew Collins ⓘ, Penelope Maher ⓘ, Stephen I. Thomson ⓘ, Geoffrey K. Vallis ⓘ

College of Engineering, Mathematics and Physical Sciences, University of Exeter, Exeter, UK

We would like to thank editor and two anonymous reviewers for their comments, which have helped us improve the manuscript. In this document, the original comments from editor and reviewers are repeated in *italic*, and our responses are in normal roman font. In addition, the line/page numbers in review comments refer to the original submitted manuscript and the line numbers in responses are for the revised manuscript.

**Response to Comments from Executive Editor (Astrid Kerkweg)**

**Comments**

*In my role as Executive editor of GMD, I would like to bring to your attention our Editorial version 1.2: https://www.geosci-model-dev.net/12/2215/2019/. This highlights some requirements of papers published in GMD, which is also available on the GMD website in the 'Manuscript Types section: http://www.geoscientific-modeldevelopment.net/submission/manuscript_types.html. In particular, please note that for your paper, the following requirement has not been met in the Discussions paper:*

- *Code must be published on a persistent public archive with a unique identifier for the exact model version described in the paper or uploaded to the supplement, unless this is impossible for reasons beyond the control of authors. All papers must include a section, at the end of the paper, entitled "Code availability". Here, either instructions for obtaining the code, or the reasons why the code is not available should be clearly stated. It is preferred for the code to be uploaded as a supplement or to be made available at a data repository with an associated DOI (digital object identifier) for the exact model version described in the paper. Alternatively, for established models, there may be an existing means of accessing the code through a particular system. In this case, there must exist a means of permanently accessing the precise model version described in the paper. In some cases, authors may prefer to put models on their own website, or to act as a point of contact for obtaining the code. Given the impermanence of websites and email addresses, this is not encouraged, and authors should consider improving the availability with a more permanent arrangement. Making code available through personal websites or via email contact to the authors is not sufficient. After the paper is accepted the model archive should be updated to include a link to the GMD paper.*

*As GitHub is not a persistent archive, please provide a persistent release for the exact source code version used for the publication in this paper. As explained in https://www.geoscientific-model-development.net/about/manuscript_types.html the preferred reference to this release is through the use of a DOI which then can be cited in the paper. For projects in GitHub a DOI for a released code version can easily be created using Zenodo, see https://guides.github.com/activities/citable-code/ for details.*

**Response:** Many thanks for your comments about the code archive problem. We are sorry that we just published the codes on Github (without a persistent DOI) during the initial submission. Currently following your suggestion, a release of SimCloud scheme codes has been archived at Zenodo: https://doi.org/10.5281/zenodo.4382536, and we hope this could meet the requirements of GMD. The DOI and corresponding reference have be added to *Code and data availability* section in the revised manuscript.

In addition, the scripts used in the manuscript to analyze data and generate figures and tables have been archived at Zenodo (https://doi.org/10.5281/zenodo.4597263) and included as part of the 'Assets'.

**Response to Comments from Anonymous Referee #1**

**General comments**

*This study proposes a simple diagnostic cloud scheme that could readily be adapted to multiple GCMs. The authors demonstrate that the cloud scheme can capture many of the basic features of observed cloud fraction and cloud radiative effects using an idealized GCM, Isca. Skill scores of the simulated results are comparable to many of the CMIP5 models, which is impressive. The proposal of the simple cloud scheme is useful, because the cloud scheme potentially helps to understand the inter-model difference in climate simulated by multiple GCMs, which has been a serious issue in climate science. The manuscript is well written. The description of the cloud scheme is sufficiently complete to allow reproduction by fellow scientists, although there is room for improvement. I recommend accepting the manuscript for publication after minor revisions.*

**Response:** Many thanks for your comments and we will reply to your specific comments one by one.

**Specific comments**

- *P1L20 "(e.g., Zelinka et al. 2017;...)" Maybe you could cite a study by Wild et al. (2019) who recently assessed cloud radiative effects. Wild et al. (2019), Climate Dynamics, 52, 4787-4812.*
  **Response:** Thank you for your suggestion and we have referred this paper in the revised manuscript (Line 20-22).

- *P2L27 "our best estimates are between..." Maybe you could mention the results of CMIP6, which is described by Zelinka et al. (2020). Zelinka et al. (2020), GRL, 47, e2019GL085782.*
  **Response:** Good point. It would be great to include the latest results from CMIP6 models. We have added a sentence to describe the results from CMIP6 models (Lines 30-32).

- *P5L122 "grid-mean relative humidity" Readers may be interested in the details of the relative humidity definition. Is it defined with respect to the liquid saturation only, or is it defined with respect to the ice saturation at temperatures below the freezing point?*
  *In addition, I suggest that the authors briefly describe how precipitation is calculated in the Isca GCM, because it may be relevant to the relative humidity distribution. I assume that the precipitation is diagnosed to remove super-saturation. Is the super-saturation with respect to liquid or ice?*
  **Response:** We would like to clarify the issues related to the relative humidity (RH) calculation. In the derivation of the relationship between cloud fraction and RH, the data sets are from ERA5 reanalysis, in which the saturated water vapor pressure is calculated over liquid and ice (see https://apps.ecmwf.int/codes/grib/param-db?id=157). But in the Isca simulations, the saturated water pressure so as the RH values are calculated over liquid only, both in the simple cloud

and in the large-scale condensation schemes. We know this may lead to some discrepancies, but the linear coefficients (and the critical RH) values are tunable, which makes the parameters obtained from reanalysis applicable to GCMs. As far as we know, it is hard to get RH reanalysis data set that is calculated over liquid only. The changes can be found in Lines 138-141.

In Isca there are two types of precipitation: convective and large-scale precipitations. The convective rainfall is diagnosed from the convection scheme, and the one we are using is the simplified Better-Miller (SBM) scheme developed by Frierson (2007). We have mentioned this in Section 3 of the manuscript. A short summary of SBM scheme can be found at https://execlim.github.io/Isca/latest/html/modules/convection_simple_betts_miller.html. The large-scale precipitation is due to the large-scale condensation, which is accomplished by adjusting the humidity in supersaturated regions to the saturated values immediately, with temperatures adjusted to reflect this condensation. The precipitation falls out immediately, but is re-evaporated below. For the large-scale condensation, the saturated water vapor pressure is computed with respect to liquid only, consistent with RH calculations in other physical processes in Isca. We have added how the precipitation is diagnosed in Isca in manuscript in Lines 301-304.

- *P7L150 "0.95 at the surface, 0.85 at 700 hPa, and 0.99 at 200 hPa" Readers may be interested in how these values are determined.*
  **Response:** The critical RH values used in Sundqvist scheme are determined by running some sensitivity tests, and they are tunable parameters for a better simulation of cloud amount and cloud radiative effect (CRE) at the top of the atmosphere (TOA). Basically the change of critical RH at 200 hPa has impact on high cloud amount and longwave CRE at the TOA, and its increase leads to the decrease in high cloud amount and longwave CRE. While the critical RH values at surface and 700 hPa mainly control the low/middle cloud amounts and shortwave CRE, and their increase result in the reduction of lower/middle cloud amounts and less negative shortwave CRE. Changes can be found in Line 161.

- *P7L154 "a and b are determined from the least squares fitting" Are the parameters a and b dependent on pressure, as shown in Fig 2(a), or are they assumed constant globally?*
  **Response:** Both *a* and *b* are dependent on pressure and derived from the same data sets to derive the single linear coefficient in Fig. 2a, and we have clarified this in the revised manuscript (Line 166).

- *P7L164 "biases in the cloud fraction and cloud radiative effect" Maybe you could also state that the biases are positive, namely overestimation, because you discuss reasons for the overestimation and how to reduce clouds in the following sentences.*
  **Response:** Yes, it would be better to present in a more specific way. The sentence (Line 176) is modified to "the cloud fraction and LW cloud radiative effect from the large scale cloud scheme are overestimated in polar regions".

- *P7L166 "there is little subgrid-scale heterogeneity of relative humidity" How does the subgrid-scale heterogeneity cause the overestimation of cloud fraction in the model simulation? It would be helpful to readers if you could describe the mechanism in more detail.*
  **Response:** The RH scheme is based on the assumption that partial clouds form if there are fluctuations of specific humidity or temperature within a grid box, even if it is under saturated when averaged over the whole grid box. In other words, when grid box mean relative humidity is above certain critical RH value, it is assumed that subgrid-scale spatial variability in the humidity fields

leads to partial cloud fraction within the grid box. The critical RH value (or equivalent linear coefficient $a$ in Eq. (1)) can reflect the subgrid scale variability (Quaas, 2012), and smaller critical RH values indicate larger spatial variability. As the linear coefficient (or equivalent critical RH) is derived from the global mean values, it may not work well in polar region, or could not reflect the spatial variability (heterogeneity) of humidity fields in polar region. In this case, for a given RH value, the cloud fraction in polar region should be smaller than in a more turbulent environment. That is to say, the cloud fraction in polar region is overestimated if the critical RH (or linear coefficient $a$) derived from a more turbulent environment is adopted. The corresponding changes are located at Lines 176-182.

- *P7L167 "The small quantity of condensation nuclei" How does the quantity of condensation nuclei cause the overestimation of cloud fraction in the model simulation? It would be helpful to readers if you could describe the mechanism in more detail.*
  **Response:** The formation of cloud droplets in a pristine atmosphere (homogeneous nucleation) is a very difficult process requiring appreciable supersaturation (Kelvin's formula, Pages 47-48, Chapter 3 of Houze, 2014), and heterogeneous nucleation is the process whereby cloud drops actually form, in which water vapor condenses on the surface of cloud nuclei instead of pure water. Thereby, if polar region has small quantity of condensation nuclei, the clouds should be hard to form in real world even though the atmosphere is close to saturation or saturated in winter. However, these microphysics processes are not included in our scheme. Instead we diagnose the cloud fraction based on RH. As pointed by Jones et al. (2004), the atmosphere is close to saturation during polar winter due to the cold temperature, which means high RH and high cloud fraction are diagnosed in our scheme. In this case, we may overestimate cloud fraction in polar region during winter seasons.

  As we do not include microphysical process in the scheme, it is better not to mention the condensate nuclei in the manuscript. Therefore, we decide to delete this sentence in the manuscript.

- *P7L174 "only polar regions will be affected" It appears to me that the freeze-dry adjustment affects clouds in low latitudes as well, because cloud fraction is smaller in Fig 6(e) than in Fig 6(d) between 30S and 30N, at around 100 hPa level.*
  **Response:** We agree with you that high clouds in low latitudes are also affected a little by this freeze-dry adjustment. The original freeze-dry method from Vavrus and Waliser (2008) is applied to low cloud only, but we find the improvement is limited if we only apply this method in lower troposphere in our simulations, so we apply it through the atmosphere in this study. In doing so, we have select the specific humidity threshold to try to ensure it only have effects in polar regions. As shown in Fig. 4 of the manuscript, this threshold works well in middle and lower troposphere, but not so well in upper atmosphere, as the specific humidity is so small (above 200 hPa) that the lines in Fig. 4 coincide with each other there. An future improvement would be only apply this freeze-dry method in low and middle troposphere (e.g., below 200 hPa).

- *P9 Figure 4 caption "The thick solid and dashed black lines are specific humidity profiles" Maybe you could add that the profiles are from the Isca simulation, and whether they are annual averages or seasonal averages.*
  **Response:** Thanks for your suggestion. We have modified the Fig. 4 caption to make it clearer.

- *P12L245 "the liquid cloud fraction varies with temperature" Maybe you could reword the term "liquid cloud fraction", because "cloud fraction" has been used in the manuscript with a different meaning, namely, the areal fraction of a grid box that is covered with cloud.*

**Response:** The "liquid cloud fraction" is reworded as "proportion of liquid cloud in a grid box". (Lines 260 and 263)

- *P12L245 "which only has an influence on the effective radius" Readers may be interested in whether the "liquid cloud fraction" affects precipitation efficiency.*
  **Response:** In our simple cloud scheme, the cloud properties are only seen by radiation scheme, so the other physical processes such as convection and large-scale condensation are not influenced by clouds directly. Instead these processes would be affected by clouds through the changes of radiation due to cloud fields.

- *P13L266 "$3 \times 10^{-4}$ g kg$^{-1}$ at 220 K and $w_{l0}$ = 0.18 g kg$^{-1}$" Readers may be interested in how these values are determined.*
  **Response:** The value $3 \times 10^{-4}$ g kg$^{-1}$ is from the SPOOKIE II project, see https://www.cfmip.org/experiments/informal-experiments (click '7. SPOOKIE Phase II' on webpage) for detail. In SPOOKIE II project, the in-cloud liquid water content is a function of height and the value above 200 hPa is specified as $3 \times 10^{-4}$ g kg$^{-1}$. Here the in-cloud water mixing ratio is specified as a function of temperature, and the $w_{l0}$ (0.18 g kg$^{-1}$ is used in this study) is tuned to match the observed cloud water path and cloud radiative effect. As shown in Table 4, taking smaller value (0.15 g kg$^{-1}$) could get a closer cloud radiative effect compared to the observation, but the energy imbalance at top of the atmosphere increases about 8 Wm$^{-2}$. In this case, we still use $w_{l0}$ = 0.18 g kg$^{-1}$ when taking all of them into account. The text changes can be found in Lines 284-287.

- *P14L283 "The sea ice data ... is averaged over all years and months" Readers may be interested in why you specified annual mean, not the monthly mean, of the sea ice distribution.*
  **Response:** Thanks for your comment. The original sea ice set up is from Thomson and Vallis (2018), and in the revised manuscript we have rerun the simulations with monthly sea ice concentration as boundary condition (Lines 298-299). The tables and figures are updated correspondingly.

- *P21L403 "over the maritime continent regions" It appears to me that the positive bias of the LW CRE is pronounced over the subtropical oceans located east of the maritime continent, as shown in Fig.12g.*
  **Response:** Yes, the LW CRE biases are pronounced over the subtropical oceans east of the maritime continents and we have corrected this in the manuscript (Line 423).

- *P31L525 "all the parameters associated with the critical relative humidity ($a_s$ or $a_t$)" It would be helpful to readers if you could suggest plausible range of the parameters ($a_s$ and $a_t$), within which we are allowed to perturb the values.*
  **Response:** We agree with you that it would be easier for users to tune parameter values ($a_s$ and $a_t$) if possible range of them are known. As we have pointed in Sect. 2.2.1, the linear coefficient $a$ and critical RH ($H_c$) follows the relationship $a = 1/(1 - H_c)$. For example, if $H_c$ is in range [0.8, 0.99], then the range of $a$ should be in [5, 100]. Users can perturb $a_s$ and $a_t$ according to this relationship.

**Technical corrections**

- *P1L4 "inversion strength" inversion height?*
  **Response:** This question is a little complicated. In this manuscript, the marine low cloud is diagnosed based on ELF, as defined in Equation (7) in the manuscript. In fact, the ELF is not

only involved in the inversion height $z_{inv}$ but also the lifting condensation level $z_{LCL}$, which can be combined together to describe the boundary layer conditions.

Specifically, as pointed by Park and Shin (2019), the term $\sqrt{z_{inv} \cdot z_{LCL}}/\Delta z_s$ can be rewritten as $z_{LCL}/\Delta z_s \cdot \sqrt{1 + (z_{inv} - z_{LCL})/z_{LCL}}$, in which $z_{LCL}/\Delta z_s$ is a proxy of surface moisture, and $(z_{inv} - z_{LCL})/z_{LCL}$ quantifies the strength of the vertical decoupling of the inversion base air from the surface. The ELF predicts that low-level cloud fraction increases as the near-surface air gets more wet (smaller $z_{LCL}$) and as the planetary boundary layer becomes more vertically coupled (smaller $z_{inv}$). Thereby we think using 'inversion height' is not enough to represent the proxy ELF.

- *P1L4 "a simple function of relative humidity" specific humidity?*
  **Response:** Fixed. Yes, it should be 'specific humidity'.

- *P6L148 "Cs=1-sqrt()" The definition of the Cs could be written as max(0, 1-sqrt()).*

  **Response:** We have rewritten the Equation (3) to $C_s = \max\left(0,\ 1 - \sqrt{\frac{1-H}{1-H_c}}\right)$ in the revised manuscript. Note that in the code we have implemented this formula correctly.

- *P19 Figure 9, units shown near the color scales g m$^{-2}$?*
  **Response:** Thanks for spotting this and the units in Fig. 9 have been updated.

- *P20L376 "Fig. 9e" Figs. 9ef?*
  **Response:** Yes, we agree with you that we can get the results described in the manuscript only when combining the Figs. 9e and 9f together. Fig. 9e shows the difference between FD and LS simulations, where the difference in marine stratocumulus regions such as Peru and California coasts is small, while Fig. 9f shows the difference between ALL and FD runs. Therefore we should use both figures together if we want to get the difference between ALL and LS simulations. We have updated Fig. 9e to Figs. 9e and 9f in the manuscript.

- *P20L384 "Fig.10c" Fig.10d?*
  **Response:** Fixed. Yes, observation should be Fig. 10d.

- *P20L393 "LD" FD?*
  **Response:** Fixed. Yes, it should be 'FD'.

- *P22L418 "Isca simulations have more weakly ascending regions and fewer weakly descending regions" Fewer weakly ascending regions and more weakly descending regions?*
  **Response:** Yes, you are right. We have corrected this sentence in the revised manuscript.

- *P35L619 "Bony, S." Bony, S. and J.-L. Dufresne?*
  **Response:** Thank you for spotting this and the bibliography has been updated.

**Response to Comments from Anonymous Referee #2**

**General comments**

*This manuscript documents a simple diagnostic cloud scheme, SimCloud, which aims to understand the dependence of cloud fraction on different diagnostic schemes and tuning parameters. The SimCloud can switch various cloud schemes within a single model framework, facilitating the evaluation of the inter-model spread in cloud biases. The authors implement the SimCloud into an idealized climate model, Isca, and perform some sensitivity experiments to show how to evaluate cloud biases in the model. Although the schemes used in the SimCloud are very simple and not new, the tool will be of importance to understand the source of uncertainties in cloud and radiation fields among multiple models for future study. The study is generally well conducted and the methods used are appropriate. I feel that this work fits within the scope of the Geoscientific Model Development. I recommend publishing this manuscript following minor revisions because the authors need to provide some additional discussion, and have several minor presentation issues to address detailed below.*

**Response:** Thank you for your general comments and we agree with them all. Now we would like to reply to your specific comments one by one.

**Specific comments**

- *Abstract: Although the abstract is supported by data and clearly reviewed the main findings of the present study, it would be helpful to readers if the authors add a brief description of the purposes of the SimCloud (e.g., Lines 529-534).*
  **Response:** Good suggestion. We have updated the abstract to describe our purpose to implement the SimCloud scheme. In general, we hope to provide a scheme with modest level of complexity and its dependence on various parameters is transparent (Line 2), so that it can be used to understand the cloud distribution and explore its possible changes (Lines 14-15). In addition, we hope it can be applied to other GCMs (Lines 8-9).

- *Line 34: The simulation of low clouds nevertheless remains problematic in GCMs: This paragraph mainly describes the bias in cloud feedback, but this paper particularly focuses on the climatological cloud biases under the present-day simulations. A brief discussion about systematic biases among GCMs (e.g., too bright low-cloud bias; Nam et al., 2012) would be beneficial to the readers.*
  **Response:** Thank you for your suggestion and we have referred this paper to describe the systematic biases of low clouds in the CMIP5 models as follows (Lines 39-42):
  "The simulation of low clouds nevertheless remains problematic in GCMs: they tend to underestimate tropical low cloud cover and to overestimate its optical depth, a systematic issue known as 'too few, too bright' problem in models from the fifth phase of the Coupled Model Intercomparison Project (CMIP5; Nam et al., 2012)."

- *Lines 83 and/or 102: More detailed description of the freeze-dry method should be provided. Or, please consider simply adding (discussed in Sect. 2.2.2) here.*
  **Response:** We have added "(discussed in Sect. 2.2.2)" to let readers easily find how 'freeze-dry' adjustment works (Line 91).

- *Line 107-108: Does the cloud scheme predict cloud droplet number concentration (i.e., two-moment scheme)? Please clarify how the model represents the Twomey effect (cloud albedo effect). Also, please consider adding a description of the cloud lifetime effect.*

**Response:** There is no microphysics scheme in our simple cloud scheme, so it does not predict cloud droplet number concentration. The simple cloud scheme only diagnose the cloud fraction, cloud water content and effective radius, which are employed by the radiation scheme (SOCRATES used here) to calculate the optical depth of clouds, the properties related to cloud albedo.

- *Line 149-150: This may need references.*
  **Response:** These critical relative humidity values are determined by sensitivity test. Please refer to our response to the same comment from Reviewer #1.

- *Line 265-266 and Eq. (12): This also may need reference(s).*
  **Response:** The equation is from the SPOOKIE II project, see https://www.cfmip.org/experiments/informal-experiments for detail. The in-cloud water content is specified as a function of height in the linked document, but in the second version of the protocol, it has been updated as a function of temperature. As the results of this project have not been published yet, we learned this equation through personal communication with Paulo Ceppi (leader of the project) and Mark Webb, and we have acknowledged them at the end of the manuscript. The parameter used in our manuscript is tunned to get a relatively realistic cloud water path compared to observation. Also see our response to Reviewer #1.

- *Line 283: Just to check, but seasonally varying means that the authors used monthly SST data, right?*
  **Response:** Yes, we use monthly SST climatology as boundary condition (see the update in Line 298).

- *Sect. 3: How does the SimCloud/Isca model treat aerosol chemistry and aerosol-cloud interactions? Adding a brief description in this section would be helpful for readers.*
  **Response:** Currently there is no aerosol chemistry and aerosol-cloud interaction in SimCloud/Isca model.

- *Line 304 and Table 3: I assume that the cloud water path is the sum of liquid and ice. Please specify the definition here.*
  **Response:** Yes, the "cloud water path" in the manuscript is referred to the sum of liquid and ice. We have clarified this in the caption of Table 3 in the revised manuscript.

- *Table 3 (and/or Fig. 7): Although the authors used cloud amount data from ISCCP, isnt it better to use CALIPSO-GOCCP data here?*
  **Response:** Thank you very much for your suggestion. We have employed CALIPSO-GOCCP dataset (2007-2015) for a new comparison and it turns out that Isca simulation results are closer CALIPSO-GOCCP than ISCCP-H (2005-2014) dataset (see Table 1). Thus we decided to use CALIPSO-GOCCP dataset for cloud amount comparison and Table 3 and Fig. 7 in revised manuscript have been updated.

- *Lines 320 and 333-335: In my experiences, cloud fraction emulated from satellite instrument simulator (lidar simulator) is fewer by approximately 5-10% than the model native output. This has also been reported in previous studies (e.g., Cesana and Chepfer, 2013). So the Isca simulations are actually close to the CALIPSO-GOCCP observations if using a simulator. Please consider adding this note here.*
  *Cesana, G., and H. Chepfer, 2013: Evaluation of the cloud thermodynamic phase in a climate*

Table 1: Cloud amount comparison (units: %)

| | ISCCP-H | CALIPSO-GOCCP | Linear_LS | Linear_FD | Linear_ALL | Sundqvist_LS | Sundqvist_FD | Sundqvist_ALL |
|---|---|---|---|---|---|---|---|---|
| Low cloud | 27.4 | **40.4** | 54.9 | 49.3 | 48.8 | 53.8 | 48.6 | 47.7 |
| Middle cloud | 21.1 | **20.3** | 25.6 | 20.6 | 20.7 | 25.2 | 20.2 | 20.0 |
| High cloud | 13.1 | **31.6** | 43.7 | 31.0 | 31.1 | 36.8 | 26.0 | 26.0 |
| Total cloud | 65.1 | **68.9** | 76.4 | 66.8 | 66.5 | 73.0 | 63.8 | 63.2 |

*model using CALIPSO-GOCCP. J. Geophys. Res. Atmos., 118, 79227937.*

**Response:** Thank you very much for this comment. As mentioned in last response, we have replaced the ISCCP data set with the CALIPSO-GOCCP product, and the cloud fraction simulated from Isca is closer to CALIPSO-GOCCP. As CALIPSO-GOCCP data set is designed to evaluate GCM cloudiness, we compare the model output with it directly. The changes can be found in Lines 349-352.

- *Figure 9: Please modify the unit in the figure ($Wm^{-2} => gm^{-2}$).*
  **Response:** Thanks for finding this typo. The units in Fig. 9 have been updated and this typo in the script we used to plot this figure has also been fixed (see here).

- *Line 371-373: I think that the overestimate of CWP can be caused by neglecting the ice- and mixed-phase microphysics scheme as well (e.g., Bergeron-Findeisen process).*
  **Response:** Thank for this comment and we have added this explanation to the manuscript in Lines 388-389.

- *Line 400-401 is closer to the observed value: The net CRE bias is more than 10 $Wm^{-2}$, so this sentence is not appropriate. This result means that more fundamental errors can also exist in microphysical properties (e.g., effective radius) and/or other processes in the model in addition to the macrophysical properties (cloud fraction).*
  **Response:** Thank you for pointing out this. We have rewritten the corresponding sentence based on your comment (Lines 419-421).

- *Line 578-580: In addition to the cloud scheme, recent studies showed that treatment of precipitation is also a very important component in GCMs. For example, some GCMs that incorporate the prognostic type of precipitation scheme (e.g., CAM, MIROC, etc.) have improved some systematic biases in the magnitude of aerosol-cloud interactions and rain formation processes with more realistic cloud and radiation fields. A brief discussion of the importance of process-based model developments against the simple model approach would be beneficial.*
  **Response:** Good suggestion. A brief discussion is added to manuscript in Lines 598-603.

**References**

Frierson, D. M.: The dynamics of idealized convection schemes and their effect on the zonally averaged tropical circulation, J. Atmos. Sci., 64, 1959–1976, https://doi.org/10.1175/jas3935.1, 2007.

Houze, R. A.: Cloud dynamics, vol. 104, Elsevier, https://doi.org/10.1016/c2009-0-18311-3, 2014.

Jones, C. G., Wyser, K., Ullerstig, A., and Willén, U.: The Rossby Centre Regional Atmospheric Climate Model Part II: Application to the Arctic climate, AMBIO: A Journal of the Human Environment, 33, 211–220, https://doi.org/10.1579/0044-7447-33.4.211, 2004.

Nam, C., Bony, S., Dufresne, J.-L., and Chepfer, H.: The 'too few, too bright' tropical low-cloud problem in CMIP5 models, Geophys. Res. Lett., 39, https://doi.org/10.1029/2012gl053421, 2012.

Park, S. and Shin, J.: Heuristic estimation of low-level cloud fraction over the globe based on a decoupling parameterization, Atmos. Chem. Phys., 19, 5635–5660, https://doi.org/10.5194/acp-19-5635-2019, 2019.

Quaas, J.: Evaluating the "critical relative humidity" as a measure of subgrid-scale variability of humidity in general circulation model cloud cover parameterizations using satellite data, J. Geophys. Res., 117, https://doi.org/10.1029/2012jd017495, 2012.

Thomson, S. I. and Vallis, G. K.: Atmospheric response to SST anomalies. Part I: Background-state dependence, teleconnections, and local effects in winter, J. Atmos. Sci., 75, 4107–4124, https://doi.org/10.1175/jas-d-17-0297.1, 2018.

Vavrus, S. and Waliser, D.: An improved parameterization for simulating Arctic cloud amount in the CCSM3 climate model, J. Climate, 21, 5673–5687, https://doi.org/10.1175/2008jcli2299.1, 2008.